# Perceptrons and Localization of Attention's Mean-Field Landscape

**Antonio Álvarez-López** [1]  **Borjan Geshkovski** [2]  **Domènec Ruiz-Balet** [3]

## Abstract

The forward pass of a Transformer can be seen as an interacting particle system on the unit sphere: time plays the role of layers, particles that of token embeddings, and the unit sphere idealizes layer normalization. In some weight settings the system can even be seen as a gradient flow for an explicit energy, and one can make sense of the infinite context length (*mean-field*) limit thanks to Wasserstein gradient flows. In this paper we study the effect of the perceptron block in this setting, and show that critical points are generically atomic and localized on subsets of the sphere.

## 1. Introduction

The distinctive operation of Transformers (Vaswani et al., 2017) is self-attention, which updates each token embedding by aggregating information from the others through a particular trainable weighted average. This mechanism is composed across depth with residual connections, layer normalization, and position-wise feed-forward blocks (typically two-layer perceptrons), producing highly structured yet poorly understood representation dynamics.

For studying signal propagation, it is useful to cast the dynamics of token representations as an interacting particle system (Geshkovski et al., 2025; Sander et al., 2022). Motivated by the perspective of treating layer depth as a continuous time variable (Chen et al., 2018) and by the fact that common normalization schemes (e.g., RMSNorm) keep embeddings on a compact manifold, one idealizes token embeddings as particles $x_i(t)$ evolving on the unit sphere $\mathbb{S}^{d-1}$ (Geshkovski et al., 2025). Self-attention then becomes a state-dependent coupling: each

particle moves toward a weighted average of the others, with weights given by a softmax kernel of pairwise inner products. This interacting-particle viewpoint—which we adopt throughout—has gained traction in the past few years. It links Transformers to nonlinear consensus and collective-dynamics models (Lu et al., 2020; Dutta et al., 2021; Geshkovski et al., 2023), and it also interfaces naturally with statistical mechanics treatments (Cowsik et al., 2025; Giorlandino & Goldt, 2025; Tiberi et al., 2024; Cui et al., 2025). While most of these works study idealized models, we emphasize that scaling laws derived by (Cowsik et al., 2025)—subsequently studied in (Chen et al., 2025a; Bruno et al., 2026; Giorlandino & Goldt, 2025)—have been used in training large language models (OLMo2 7B & 13B, (OLMo Team et al., 2025)).

One object of interest in modern applications is very large context lengths. This is precisely where the *mean-field limit* of the interacting particle system becomes relevant: as the number of tokens $n$ grows, the empirical measure of particles converges to a measure solving a nonlinear partial differential equation on the sphere whose velocity field is the attention interaction. One can justify this mean-field limit by classical arguments (Geshkovski et al., 2025). A convenient advantage of the particle perspective is the transparent variational structure: the finite-$n$ particle dynamics can be written as a preconditioned (or weighted-metric) gradient flow of an interaction energy, yielding gradient *descent* when the key-query-value-induced quadratic form is positive semidefinite and gradient *ascent* otherwise (Geshkovski et al., 2025). This gradient flow structure persists in the mean-field limit, where the limiting PDE is a (weighted) Wasserstein gradient flow (Jordan et al., 1998; Otto, 2001). This framework naturally connects to the study of equilibrium measures for nonlocal interaction energies, including the structural properties of their minimizers (Cañizo et al., 2015; Carrillo et al., 2017).

Exploiting this structure has already enabled a rigorous analysis of the *ascent* (attractive) regime, where attention concentrates and synchronized clusters emerge, as well as of the *descent* (repulsive) regime, where particles tend to equidistribute uniformly (Geshkovski et al., 2023; 2025; Chen et al., 2025b; Criscitiello et al., 2024; Polyanskiy et al., 2025; Castin et al., 2025; Yu et al., 2024; Alcalde et al., 2025a). These results have been extended to settings with very gen-

[1]Departamento de Matemáticas, Universidad Autónoma de Madrid, 28049 Madrid, Spain [2]Inria & Laboratoire Jacques-Louis Lions, Sorbonne Université, 75005 Paris, France [3]Departament de Matemàtiques, Universitat de Barcelona, 08007 Barcelona, Spain. Correspondence to: Antonio Álvarez-López .

*Proceedings of the 43rd International Conference on Machine Learning*, Seoul, South Korea. PMLR 306, 2026. Copyright 2026 by the author(s).

eral trained weights (Abella et al., 2025; Burger et al., 2025; Koubbi et al., 2024), to decoder-only architectures with causal masking (Karagodin et al., 2024; Duerinckx et al., 2026; Barbero et al., 2025), to finer landscape phenomena such as metastability (Geshkovski et al., 2024a; Bruno et al., 2025a;b; Alcalde et al., 2025b; Altafini, 2025), to diffusive variants (Gerber et al., 2025; Balasubramanian et al., 2025; Shalova & Schlichting, 2026; Peletier & Shalova, 2025), and to stochastic scaling limits arising from random weights (Fedorov et al., 2026; Koubbi et al., 2026; Agazzi et al., 2026).

A key architectural ingredient is still largely absent from the above variational mean-field picture: the feed-forward perceptron block. We emphasize that attention-only mean-field limits admit non-atomic stationary densities in repulsive regimes—indeed, the uniform distribution on the sphere is the global minimizer of the resulting energy! Practical evidence also suggests that perceptrons qualitatively change the geometry by counteracting attention-driven collapse, inducing an order–chaos transition (Cowsik et al., 2025). From the PDE viewpoint, the perceptron appears as an external drift, and we argue that it indeed reshapes the landscape, but not necessarily countering the above-cited studies: even when pure attention permits continuous equilibria, the stationary states of the coupled dynamics are generically discrete or singular.

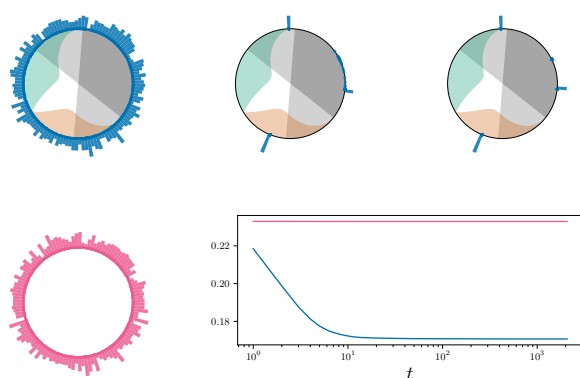

*Figure 1.* Gradient descent with ReLU perceptron in $d = 2$, starting from the uniform measure, with $\beta = 1$. **Top:** particle histograms for the dynamics at initial, intermediate and final times. **Bottom left:** final configuration for pure self-attention without a perceptron. **Bottom right:** the energy $\mathsf{E}_{\beta,\vartheta}$ with (blue) and without (pink) the perceptron. Background shading shows the perceptron landscape (green: $> 0$; orange: $< 0$). The background shading differentiates the active regions: areas where $a_j \cdot x + b_j > 0$ for some $j$ are shown in lighter gray, while the darkest gray regions correspond to the "dead zones" where the potential vanishes. Animation available at https://github.com/antonioalvarezl/2026-MLP-Attention-Energy/blob/main/examples/USAd.gif

.

## 1.1. Our contributions

We study mean-field attention as a Wasserstein gradient flow for an energy that couples (unnormalized) attention interactions with a perceptron-induced potential. Our main contributions are:

- For ReLU perceptrons in $d = 2$, every stationary measure has finite support (**Theorem 3.1**); for analytic activations (e.g., GeLU), the same holds for "stable stationary measures" (**Theorem 3.2**). In $d \geqslant 2$, stationary measures are necessarily singular, and for a dense open set of parameters they are purely atomic (**Theorem 3.3**).

- In the descent (thus repulsive) regime, although the perceptron induces discreteness, we show an anti-concentration phenomenon: the mass of any cluster at the interaction scale $1/\sqrt{\beta}$ is bounded by a numerical constant (**Theorem 3.5**). Consequently, the limit measure (provided it exists) is atomic but cannot collapse to a single atom, in contrast to the attractive regime. We also show that the number of heavy atoms scales with $\sqrt{\beta}$ (**Corollary 3.6**).

- We characterize the extremal points of the energy in **Proposition 3.7**. For ReLU perceptrons, maximizers reduce to a finite family of quadratic programs, yielding explicit solutions in some cases.

---

**Model at a glance.** We model a context of many tokens as a probability measure $\mu \in \mathcal{P}(\mathbb{S}^{d-1})$.

*Unnormalized self-attention* is the velocity field

$$\nabla \frac{\delta \mathsf{E}_\beta}{\delta \mu}[\mu](x) = \int e^{\beta x \cdot y} \, \mathbf{P}_x^\perp \, y \, \mathrm{d}\mu(y),$$

where

$$\mathsf{E}_\beta[\mu] := \frac{1}{2\beta} \iint e^{\beta x \cdot y} \, \mathrm{d}\mu(x) \, \mathrm{d}\mu(y).$$

*Perceptron:* a drift $\mathsf{u}_\vartheta(x)$ is added to the velocity; in the conservative, ReLU case $\mathsf{u}_\vartheta = \frac{1}{2}\nabla \mathsf{v}_\vartheta$ with $\mathsf{v}_\vartheta(x) = \sum_j \omega_j (a_j \cdot x)_+^2$. The resulting mean-field dynamics are

$$\partial_t \mu(t) \pm \operatorname{div}\left(\mu(t)\left(\nabla \frac{\delta \mathsf{E}_\beta}{\delta \mu}[\mu(t)] + \mathsf{u}_\vartheta\right)\right) = 0,$$

and are a Wasserstein gradient flow (ascent+/descent−) for

$$\mathsf{E}_{\beta,\vartheta}[\mu] := \mathsf{E}_\beta[\mu] + \frac{1}{2}\int \mathsf{v}_\vartheta(x) \, \mathrm{d}\mu(x).$$

*Stationary measures* satisfy $\nabla \frac{\delta \mathsf{E}_{\beta,\vartheta}}{\delta \mu}[\mu] = 0$ on $\operatorname{supp} \mu$.

**Takeaway:** adding the perceptron localizes the landscape; generically, stationary states are finitely supported atomic measures, even in regimes where pure repulsive attention admits diffuse equilibria.

---

**Conflict of Interest Disclosure.** The authors declare no financial conflicts of interest related to this work.

## 2. Setup

Let $\mathcal{P}(\mathbb{S}^{d-1})$ denote the space of Borel probability measures on $\mathbb{S}^{d-1}$ and $\sigma_d$ the uniform measure on $\mathbb{S}^{d-1}$. For a smooth $f : \mathbb{S}^{d-1} \to \mathbb{R}$ and $x \in \mathbb{S}^{d-1}$, the spherical gradient is

$$\nabla f(x) := \mathbf{P}_x^\perp \nabla_{\mathbb{R}^d} \tilde{f}(x)$$

where $\mathbf{P}_x^\perp := I_d - xx^\top$ is the orthogonal projection onto $\mathsf{T}_x\mathbb{S}^{d-1}$ and $\tilde{f} : \mathbb{R}^d \to \mathbb{R}$ is a(ny) smooth extension of $f$; $-\mathrm{div}$ denotes the adjoint of $\nabla$. All integrals are taken over $\mathbb{S}^{d-1}$, which we drop to ease reading.

### 2.1. Wasserstein gradient flows

We study the evolution of measures driven by a functional $\mathsf{E} : \mathcal{P}(\mathbb{S}^{d-1}) \to \mathbb{R}$. A curve $(\mu(t))_{t \geqslant 0} \subset \mathcal{P}(\mathbb{S}^{d-1})$ is a *Wasserstein gradient flow*—in the steepest descent convention—for $\mathsf{E}$ if it is locally absolutely continuous in $(\mathcal{P}(\mathbb{S}^{d-1}), W_2)$ and there exists a velocity field $v(t) \in L^2(\mu(t); \mathsf{T}\mathbb{S}^{d-1})$ such that

$$\partial_t \mu(t) + \mathrm{div}\,(\mu(t)v(t)) = 0, \qquad (2.1)$$

in the sense of distributions $\mathcal{D}'(\mathbb{R}_{\geqslant 0} \times \mathbb{S}^{d-1})$, with velocity given by the gradient of the first variation,

$$v(t) = -\nabla \frac{\delta \mathsf{E}}{\delta \mu}[\mu(t)].$$

We recall that the first variation $\frac{\delta \mathsf{E}}{\delta \mu}[\mu]$ is defined (up to an additive constant) by the relation

$$\frac{\mathrm{d}}{\mathrm{d}\varepsilon} \mathsf{E}(\mu + \varepsilon\chi)\Big|_{\varepsilon=0} = \int \frac{\delta \mathsf{E}}{\delta \mu}[\mu]\,\mathrm{d}\chi,$$

for signed measures $\chi$ with $\chi(\mathbb{S}^{d-1}) = 0$ and such that $\mu + \varepsilon\chi \in \mathcal{P}(\mathbb{S}^{d-1})$ for $|\varepsilon|$ small. In the Wasserstein geometry, the relevant perturbations are transport ones $\chi = -\mathrm{div}(\mu\xi)$ with $\xi \in L^2(\mu; \mathsf{T}\mathbb{S}^{d-1})$, which identifies the Wasserstein gradient as above; see (Ambrosio et al., 2008).

### 2.2. Critical points

We are primarily interested in the asymptotic behavior of these flows, which relates to the critical points of $\mathsf{E}$. A measure $\mu \in \mathcal{P}(\mathbb{S}^{d-1})$ is a *critical point* of $\mathsf{E}$, or stationary solution to (2.1), if

$$\nabla \frac{\delta \mathsf{E}}{\delta \mu}[\mu](x) = 0 \qquad \text{for } \mu\text{-a.e. } x. \qquad (2.2)$$

We are also interested in second-order Wasserstein critical points. Recall that

$$\mathsf{T}_\mu \mathcal{P}(\mathbb{S}^{d-1}) := \overline{\{\nabla\phi : \phi \in C^\infty(\mathbb{S}^{d-1})\}}^{L^2(\mu)} \qquad (2.3)$$

is the tangent space at $\mu$, a Hilbert subspace of $L^2(\mu; \mathsf{T}\mathbb{S}^{d-1})$. Suppose $\mu$ lies in the regular (Riemannian) part of $(\mathcal{P}_2(\mathbb{S}^{d-1}), W_2)$ and $\mathsf{E}$ is twice differentiable in the $W_2$-sense at $\mu$ (Villani, 2009, Ch. 15). Then the *Wasserstein Hessian* at $\mu$ in the direction $\xi \in \mathsf{T}_\mu \mathcal{P}(\mathbb{S}^{d-1})$ is defined by

$$\mathrm{Hess}_\mu \mathsf{E}(\xi, \xi) := \frac{\mathrm{d}^2}{\mathrm{d}t^2}\Big|_{t=0} \mathsf{E}(\mu(t)),$$

where $(\mu(t))$ is the $W_2$-geodesic emanating from $\mu$ with initial velocity field represented by $\xi = \nabla\phi$, i.e. (for $|t|$ small) $\mu(t) = (T(t))_{\#}\mu$ with $T(t)(x) = \exp_x(t\nabla\phi(x))$.[1]

A critical point $\mu$ is *second-order positive-definite* (SOPD) if the Wasserstein Hessian at $\mu$ is well-defined and

$$\mathrm{Hess}_\mu \mathsf{E}(\xi, \xi) \geqslant 0 \qquad \text{for all } \xi \in \mathsf{T}_\mu \mathcal{P}(\mathbb{S}^{d-1}), \qquad (2.4)$$

and it is *strictly* SOPD if, moreover, there is $\kappa > 0$ such that

$$\mathrm{Hess}_\mu \mathsf{E}(\xi, \xi) \geqslant \kappa \|\xi\|_{L^2(\mu)}^2. \qquad (2.5)$$

A SOPD critical point is *degenerate* if it is not strictly SOPD, i.e. if there exists $\xi \neq 0$ with $\mathrm{Hess}_\mu \mathsf{E}(\xi, \xi) = 0$. If (2.4) fails (there exists $\xi$ with $\mathrm{Hess}_\mu \mathsf{E}(\xi, \xi) < 0$), we call $\mu$ unstable (a saddle/maximum direction).

We focus on SOPD Wasserstein critical points because they ought to capture the appropriate notion of *stable* equilibria for Wasserstein gradient (descent) flows and because quantitative convergence estimates are typically driven by second-order information (e.g. PL-type inequalities) around such points (Villani, 2009; Otto, 2001; Chen et al., 2025b). In our coupled attention–perceptron setting, this perspective also highlights a surprising phenomenon: even in the repulsive regime, stable stationary states are forced to be discrete, so the mean-field flow can evolve from a smooth initialization toward a genuinely clustered configuration rather than a diffuse equilibrium.

### 2.3. Mean-field formulation of Transformers

As in (Geshkovski et al., 2025), we model the layer-wise evolution of token embeddings as the flow of an interacting particle system on $\mathbb{S}^{d-1}$, where "time" indexes the layers of the architecture. At the mean-field level, the distribution of a particle under pure self-attention dynamics follows the Wasserstein gradient flow of

$$\mathsf{E}_\beta[\mu] := \frac{1}{2\beta} \iint e^{\beta x \cdot y}\,\mathrm{d}\mu(x)\,\mathrm{d}\mu(y).$$

Its first variation is

$$\frac{\delta \mathsf{E}_\beta}{\delta \mu}[\mu](x) = \frac{1}{\beta} \int e^{\beta x \cdot y}\,\mathrm{d}\mu(y),$$

---

[1] Here $\exp_x$ denotes the exponential map on $\mathbb{S}^{d-1}$. In general (e.g. for singular $\mu$), geodesics need not be uniquely determined by an $L^2(\mu)$ velocity field; in that case one may instead work with directional second variations along Monge perturbations, but we will not need this level of generality here.

and the Wasserstein gradient in (2.1) reads

$$\nabla \frac{\delta \mathsf{E}_\beta}{\delta \mu}[\mu](x) = \mathbf{P}_x^\perp \int e^{\beta x \cdot y} y \, \mathrm{d}\mu(y).$$

This setting corresponds to the toy model analyzed in (Geshkovski et al., 2025) in which the key-query weights $K, Q$ and value weights $V$ are constant in time and satisfy $B := Q^\top K = V = I_d$. The framework and results can however be generalized to very general weights (Abella et al., 2025; Geshkovski et al., 2025; Karbevski & Mijoski, 2025)—see also Section 2.4 below.

As discussed in (Geshkovski et al., 2025), the perceptron component can be incorporated at the mean-field level as an additional drift field, yielding

$$\partial_t \mu(t) \pm \mathrm{div}\left(\mu(t)\left(\nabla \frac{\delta \mathsf{E}_\beta}{\delta \mu}[\mu(t)] + \mathsf{u}_\vartheta\right)\right) = 0, \quad (2.6)$$

where

$$\mathsf{u}_\vartheta(x) := \mathbf{P}_x^\perp \sum_{j \in [\![1,d]\!]} \boldsymbol{\omega}_j \sigma\left(a_j \cdot x + b_j\right),$$

with weights $\vartheta = (a_j, \boldsymbol{\omega}_j, b_j)_{j \in [\![1,d]\!]}$, where $a_j, \boldsymbol{\omega}_j \in \mathbb{R}^d$ and $b_j \in \mathbb{R}$. We refer to the $+$ case in (2.6) as *ascent*, and $-$ as *descent*. One typically uses the ReLU or the GeLU, defined respectively as

$$\sigma(s) = s_+ \quad \text{and} \quad \sigma(s) = \frac{s}{2}\left(1 + \mathrm{erf}\left(\frac{s}{\sqrt{2}}\right)\right).$$

Our analysis distinguishes between these two cases. For simplicity, we henceforth omit the biases $b_j$, discussing their inclusion later in Remark 3.4.

When $\mathsf{u}_\vartheta$ derives from a scalar potential, the coupled dynamics remain a Wasserstein gradient flow. This holds if and only if the output weights $\boldsymbol{\omega}_j$ are collinear with the input weights $a_j$, meaning $\boldsymbol{\omega}_j = \omega_j a_j$ for some scalars $\omega_j \in \mathbb{R}$. Henceforth, we assume this is the case and denote the parameters by $\vartheta = (a_j, \omega_j)_{j \in [\![1,d]\!]} \in (\mathbb{R}^{d+1})^d$.

Under this condition, we can choose a primitive $\varphi$ with $\varphi'(s) = 2\sigma(s)$ and set

$$\mathsf{v}_\vartheta(x) := \sum_{j \in [\![1,d]\!]} \omega_j \varphi\left(a_j \cdot x\right), \quad (2.7)$$

so that $\frac{1}{2}\nabla \mathsf{v}_\vartheta(x) = \mathsf{u}_\vartheta(x)$. The full energy is then

$$\mathsf{E}_{\beta,\vartheta}[\mu] := \frac{1}{2\beta} \iint e^{\beta x \cdot y} \, \mathrm{d}\mu(x) \, \mathrm{d}\mu(y) + \frac{1}{2} \int \mathsf{v}_\vartheta(x) \, \mathrm{d}\mu(x),$$

and (2.6) is precisely its Wasserstein gradient flow, since

$$\nabla \frac{\delta \mathsf{E}_{\beta,\vartheta}}{\delta \mu}[\mu](x) = \int e^{\beta x \cdot y} \mathbf{P}_x^\perp y \, \mathrm{d}\mu(y) + \mathsf{u}_\vartheta(x).$$

Whenever $\sigma$ is continuous, (2.2) extends to the entire support:

$$\nabla \frac{\delta \mathsf{E}_{\beta,\vartheta}}{\delta \mu}[\mu](x) = 0 \qquad \text{for all } x \in \mathrm{supp}\,\mu. \quad (2.8)$$

## 2.4. Practical considerations

While our toy model captures the core interaction dynamics, we distinguish three differences with respect to Transformers used in real-world applications.

**Perceptrons** The drift given by the perceptron is a gradient field only under the specific weight symmetries assumed in (2.7). More general weights $\boldsymbol{\omega}_j$ could be interpreted as preconditioners, much like what is discussed in Section 2.2 in (Alcalde et al., 2025b).

**Self-attention** *(i)* Practical implementations replace $\beta x \cdot y$ with a general bilinear form $x^\top Q^\top K y$ involving query ($Q$) and key ($K$) matrices. For technical clarity, we work in the isotropic case $Q^\top K = \beta I_d$ with $V = \pm I_d$, but our main results can be extended to the setting where $Q^\top K$ is symmetric and invertible; see Remark 3.4 and the discussion after Lemma A.1. The sign of $V$ determines whether the dynamics follow gradient ascent or descent, but this does not change the stationary points.

*(ii)* In practice, attention scores are also normalized via a softmax as

$$\nabla \log \int e^{\beta x \cdot y} \, \mathrm{d}\mu(y) = \frac{1}{\frac{\delta \mathsf{E}_\beta}{\delta \mu}[\mu](x)} \nabla \frac{\delta \mathsf{E}_\beta}{\delta \mu}[\mu](x).$$

This field corresponds to a weighted Wasserstein gradient $\mathbb{W}$ obtained by rescaling the standard variation by the strictly positive weight $\mathsf{w}[\mu](x) := \frac{\delta \mathsf{E}_\beta}{\delta \mu}[\mu](x)$. The resulting dynamics are

$$\partial_t \mu(t) + \mathrm{div}\left(\mu(t)\left(\mathbb{W}\mathsf{E}_\beta[\mu(t)] + \mathsf{u}_\vartheta\right)\right) = 0, \quad (2.9)$$

where $\mathbb{W}\mathsf{E} := \frac{1}{\mathsf{w}[\mu]} \nabla \frac{\delta \mathsf{E}}{\delta \mu}[\mu]$. They retain the structure of a gradient flow (under a conformally equivalent metric). Since $\mathsf{w} > 0$, the stationarity condition (2.8) becomes:

$$\nabla \frac{\delta \mathsf{E}_\beta}{\delta \mu}[\mu] + \mathsf{w}[\mu]\mathsf{u}_\vartheta = 0 \qquad \text{on } \mathrm{supp}\,\mu. \quad (2.10)$$

To facilitate the analysis, we focus on unnormalized attention in the main text; the analogous results for (2.9), which are qualitatively similar, are detailed in **Appendix B**.

## 3. Results

We now state our main results, which characterize the stationary configurations of the mean-field dynamics (2.6).

### 3.1. Atomicity of critical points

We begin with the simplest case of the ReLU perceptron in $d = 2$. In this setting, the non-analyticity of the potential forces any stationary measure to have finite support.

**Theorem 3.1.** *Let $d = 2$, $\beta > 0$, and fix $\sigma(s) = s_+$. Assume that the weights $\vartheta$ are such that $v_\vartheta$ is not real-analytic[2] on $\mathbb{S}^1$. Then any $\mu \in \mathcal{P}(\mathbb{S}^1)$ satisfying (2.8) is purely atomic and has finite support.*

When the activation is real-analytic (e.g. GeLU), simply looking at solutions of (2.8) is not enough due to the analyticity of the potential. However, the same conclusions hold for strict SOPD Wasserstein critical points.

**Theorem 3.2.** *Let $d = 2$, $\beta > 0$. Fix weights $\vartheta$ and a real-analytic function $\sigma : \mathbb{R} \to \mathbb{R}$. If $\mu \in \mathcal{P}(\mathbb{S}^1)$ is a strict SOPD Wasserstein critical point of $\mathsf{E}_{\beta,\vartheta}$ in the sense of (2.5), then $\mu$ is purely atomic and has finite support.*

*In other words, if a SOPD Wasserstein critical point has infinite support, then it must be degenerate.*

In higher dimensions the landscape is more complicated, but the following theorem establishes that stationary measures are always singular and atomicity remains generic.

**Theorem 3.3.** *Let $d \geqslant 2$ and $\mu \in \mathcal{P}(\mathbb{S}^{d-1})$.*

*(i) Fix $\beta > 0$. If $\mu$ is stationary for $\sigma(s) = s_+$ and $\vartheta$ is such that $v_\vartheta$ is not real-analytic, or $\mu$ is a strict SOPD critical point for a real-analytic $\sigma$, then $\sigma_d(\operatorname{supp}\mu) = 0$. In particular, $\mu$ is singular with respect to $\sigma_d$.*

*(ii) If $\sigma$ is real-analytic and $\sigma(s) \neq 0$ for $s \neq 0$, then there exists an open and dense set $U_\mu \subset \mathbb{R}_{>0} \times (\mathbb{R}^{d+1})^d$ such that, if $(\beta, \vartheta) \in U_\mu$ and $\mu$ is stationary with respect to $(\beta, \vartheta)$, then $\mu$ is purely atomic with finite support.*

*(iii) If $\sigma(s) = s_+$, then there exists a dense set $U_\mu \subset \mathbb{R}_{>0} \times (\mathbb{R}^{d+1})^d$ such that, if $(\beta, \vartheta) \in U_\mu$ and $\mu$ is stationary with respect to $(\beta, \vartheta)$, then the restriction of $\mu$ to the active regions $\bigcup_j \{x : a_j \cdot x > 0\}$ is purely atomic with at most countably many atoms.*

*Remark* 3.4. The results above easily extend to:

*(i)* biases $a_j \cdot x + b_j$ inside the perceptron, provided at least one of the hyperplanes $\{x : a_j \cdot x + b_j = 0\}$ intersects $\mathbb{S}^{d-1}$ in more than one point (i.e., $|b_j| < \|a_j\|$ for some $j$).

*(ii)* symmetric invertible $B$ instead of $\beta I_d$ in the interaction energy; see Remarks A.2 and B.2.

### 3.2. Anti-concentration bound

While stationary measures have a discrete nature in the repulsive case, the strict concavity of the kernel[3] $\theta \mapsto e^{\beta \cos \theta}$

---

[2]This discards the pathological cases where $v_\vartheta \equiv 0$, or specific weight symmetries for which $v_\vartheta$ effectively becomes a quadratic trigonometric polynomial.

[3]This has also been used in (Geshkovski et al., 2025; 2024c) in related but different contexts.

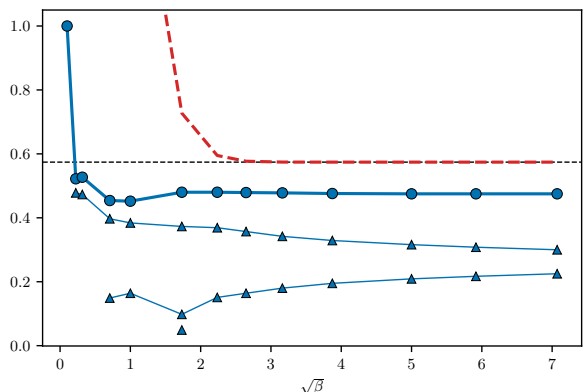

*Figure 2.* Cluster masses (in blue, the largest being the thickest) at final time across $\sqrt{\beta}$ for gradient descent with GeLU perceptron, initialized with $N = 1000$ points of mass $10^{-3}$ (see Section 4 for setup). The horizontal and red dashed lines represent the numerical term and the full upper bound in (3.3), respectively.

on $(-\beta^{-\frac{1}{2}}, \beta^{-\frac{1}{2}})$ for $\beta \to \infty$ ensures that they cannot be too concentrated.

**Theorem 3.5.** *Let $d = 2$, $\beta > 0$ and $\sigma$ globally Lipschitz with constant 1 and $\sigma(0) = 0$ for simplicity. Consider any SOPD Wasserstein critical point of the form*

$$\mu = \sum_{i \in [\![1,N]\!]} m_i \delta_{\theta_i} \in \mathcal{P}(\mathbb{S}^1) \tag{3.1}$$

*where $N \geqslant 2$, the weights $m_i > 0$ sum to 1, and the points $\theta_i \in [0, 2\pi)$ are pairwise distinct.*

*Suppose there exists a subset $\mathcal{S} \subseteq [\![1, N]\!]$ with $n = |\mathcal{S}| \geqslant 2$ satisfying the cluster condition*

$$\max_{i,j \in \mathcal{S}} \min_{k \in \mathbb{Z}} |\theta_i - \theta_j + 2\pi k| \leqslant \frac{1}{2\sqrt{\beta}}. \tag{3.2}$$

*Then, for $\beta$ sufficiently large, the total mass of this cluster satisfies*

$$\sum_{i \in \mathcal{S}} m_i \leqslant 0.5742 + O\left(e^{-\beta}\right). \tag{3.3}$$

*Moreover, if $\vartheta = (\omega_j, a_j)_j$ satisfies*

$$|\omega_1| \cdot \|a_1\|^2 + |\omega_2| \cdot \|a_2\|^2 < 0.16547, \tag{3.4}$$

*then $\mathcal{S} = [\![1, N]\!]$ satisfying (3.2) cannot hold, for any $\beta > 0$.*

It ensues that

**Corollary 3.6.** *Let $\mu$ be as in Theorem 3.5. Suppose that*

$$\operatorname{supp}\mu \subset \bigcup_{j \in [\![1,M]\!]} I_j, \qquad \text{where } \max_{j \in [\![1,M]\!]} |I_j| =: L < 2\pi,$$

*for some integer $M \geqslant 1$. Let $N_\varepsilon$ be the number of atoms with mass $\geqslant \varepsilon$ for $\varepsilon > 0$. Then, for $\beta$ large enough,*

$$N_\varepsilon \leqslant \frac{M}{\varepsilon}\left(1 + 2L\sqrt{\beta}\right)\left(0.5742 + O\left(e^{-\beta}\right)\right).$$

We finish with the natural question of *extremal* points.

**Proposition 3.7.** *Let $d \geqslant 2$, $\beta > 0$, $\vartheta = (a_j, \omega_j)_{j \in [\![1,d]\!]}$ and $\varphi' = 2\sigma$.*

*(i) The global maximizers of $\mathsf{E}_{\beta,\vartheta}$ are exactly those $\mu = \delta_x$ with*

$$x \in \arg\max_{y \in \mathbb{S}^{d-1}} \sum_{j \in [\![1,d]\!]} \omega_j \varphi(a_j \cdot y).$$

*(ii) $\mathsf{E}_{\beta,\vartheta}$ has a unique global minimizer $\mu_*$ which is invariant under rotations that fix every $a_j$ such that $\omega_j \neq 0$.*

### 3.3. Maximizers for ReLU perceptrons

We compute the maximizers in Proposition 3.7 for $\sigma(s) = s_+$. Define

$$\mathscr{Z} := \{x \in \mathbb{S}^{d-1} : a_j \cdot x = 0 \text{ for some } j \in [\![1,d]\!]\} \quad (3.5)$$

and let $I$ be any connected component of $\mathbb{S}^{d-1} \setminus \mathscr{Z}$ with active index set

$$J_I := \{j \in [\![1,d]\!] : a_j \cdot x > 0 \text{ for all } x \in I\}. \quad (3.6)$$

The signs of $x \mapsto a_j \cdot x$ are fixed on $I$, hence

$$\mathsf{v}_\vartheta(x) = x^\top B_I x, \qquad B_I := \sum_{j \in J_I} \omega_j\, a_j a_j^\top \quad .$$

Thus Proposition 3.7 reduces to solving and comparing a finite collection of constrained quadratic programs:

$$\max_{x \in \mathbb{S}^{d-1}} \mathsf{v}_\vartheta(x) = \max_\alpha \max_{x \in \overline{I}} x^\top B_I x. \quad (3.7)$$

This yields finitely many candidates, one for each connected component of $\mathbb{S}^{d-1} \setminus \mathscr{Z}$. On each $\overline{I}$, either a principal eigenvector of $B_I$ that lies in $I$ maximizes $x^\top B_I x$, or else all maximizers lie on $\partial I$.

Across regions, the maximizers of $\mathsf{v}_\vartheta$ may form a singleton, a finite set, or a continuum. In special cases they can be described explicitly:

**(i) Collinear directions.** Suppose $a_j = \alpha_j a$ with $a \in \mathbb{R}^d$ and $\alpha_j \in \mathbb{R}$. Then

$$\mathsf{v}_\vartheta(x) = (a \cdot x)_+^2 \sum_{\alpha_j > 0} \omega_j \alpha_j^2 + (a \cdot x)_-^2 \sum_{\alpha_j < 0} \omega_j \alpha_j^2.$$

Hence

$$\max_{x \in \mathbb{S}^{d-1}} \mathsf{v}_\vartheta(x) = \|a\|^2 \max\left\{\sum_{\alpha_j > 0} \omega_j \alpha_j^2,\ \sum_{\alpha_j < 0} \omega_j \alpha_j^2,\ 0\right\}.$$

If $\max \mathsf{v}_\vartheta > 0$, the maximizer is $\delta_{a/\|a\|}$ or $\delta_{-a/\|a\|}$, depending on which sum is larger; both are maximizers in the tie case. Otherwise, any $\delta_x$ with $a \cdot x = 0$ is a maximizer.

**(ii) Diagonal directions.** If $a_j = \alpha_j e_j$ with $\alpha_j \in \mathbb{R}$, and $\omega_j \equiv 1$, then

$$\max_{x \in \mathbb{S}^{d-1}} \mathsf{v}_\vartheta(x) = \max_{x \in \mathbb{S}^{d-1}} \sum_{j \in [\![1,d]\!]} (\alpha_j x_j)_+^2 = \max_{j \in [\![1,d]\!]} |\alpha_j|^2,$$

and the maximizers are those $\delta_x$ with $x_k = 0$ if $|\alpha_k| < \max_j |\alpha_j|$ and $\alpha_k x_k \geqslant 0$ if $|\alpha_k| = \max_j |\alpha_j|$. If the $\max_j$ is attained at a unique index $k$, then $x = \text{sign}(\alpha_k)e_k$.

**(iii) Nonnegative.** Assume $a_j \geqslant 0$ entrywise and $\omega_j \equiv 1$. For any $x \in \mathbb{S}^{d-1}$,

$$\sum_{j \in [\![1,d]\!]} (a_j \cdot x)_+^2 \leqslant \sum_{j \in [\![1,d]\!]} (a_j \cdot x)^2 = \|Ax\|_2^2 \leqslant \|A\|_2^2,$$

where $A$ is the matrix with rows $a_j$. By Perron–Frobenius on the entrywise nonnegative $A^\top A$, we may choose a top right singular vector $v_1 \geqslant 0$, ensuring $Av_1 \geqslant 0$. Thus $x = v_1$ saturates both inequalities, so $\delta_{v_1/\|v_1\|}$ is a maximizer (unique if the top singular value is simple).

### 3.4. Discussion

*(i) Domain geometry.* The analysis is conducted on $\mathbb{S}^{d-1}$. Despite the unique-continuation arguments naturally extend to other real-analytic compact manifolds, ruling out continuous measures globally (as in Theorem 3.1 and Theorem 3.3) heavily relies on the spectral inversion of the isotropic interaction kernel $e^{\beta x \cdot y}$ via spherical harmonics (Lemma A.1). Extending this contradiction mechanism to general domains remains an open problem.

*(ii) Non-conservative drifts.* When the assumption of collinear weights is dropped, the MLP drift is no longer a gradient. The stationary condition is the weaker transport equation $\text{div}(\mu v) = 0$, rather than the pointwise condition of (2.8). Handling this generalized condition falls outside the scope of our current techniques.

*(iii) General attention kernels.* The atomicity results require $\mu \mapsto \int K(\cdot, y)\,d\mu(y)$ to be real-analytic and injective (or provide a comparable spectral inversion). On the other hand, the quantitative anti-concentration bounds in Theorem 3.5 depend on the specific local concavity of $e^{\beta \cos \theta}$. Extending these conclusions to completely general attention kernels requires corresponding analytic and local-curvature properties.

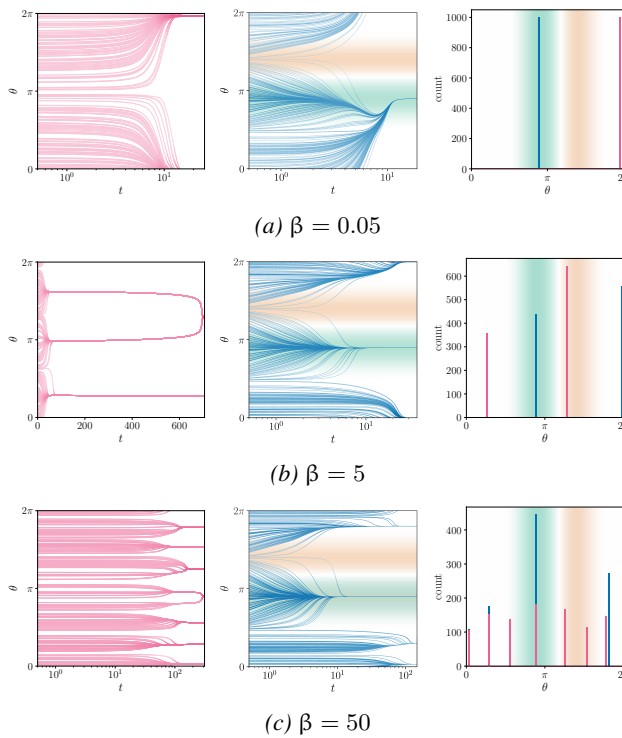

*(a)* β = 0.05

*(b)* β = 5

*(c)* β = 50

*Figure 3.* Gradient ascent on $\mathbb{S}^1$ with ReLU perceptron. **Left:** pure self-attention. **Middle:** self-attention with a ReLU perceptron. **Right:** measure at final time. Background shading represents the potential landscape (green: positive; orange: negative values).

# 4. Simulations

We complement our theory with simulations[4].

**Methodology**

Unless stated otherwise, we simulate (2.6)–(2.9) on $\mathbb{S}^1$ using 1000 particles initialized uniformly on $[0, 2\pi)$ using a fixed random seed. We use an explicit Euler scheme with $\Delta t = 0.1$. We sweep the inverse temperatures

$$\beta \in \{0.01, 0.05, 0.1, 0.5, 1, 3, 5, 7, 10, 15, 25, 35, 50\}.$$

The perceptron weights $\vartheta$ are sampled from a standard normal distribution and fixed across all runs.

For a given β, motivated by Lemma A.3, we identify clusters on $\mathbb{S}^1$ by grouping particles whose pairwise geodesic distance is at most $\min\{1/(2\sqrt{\beta}), \pi/4\}$.

The simulation stops once the cluster count remains constant over a window of 5 snapshots (with a snapshot every 10 steps) and $\max_i |\dot{\theta}_i| \leqslant 10^{-4}$. Metastability can occur (Geshkovski et al., 2024a; Bruno et al., 2025a;b): the dynamics may enter very slow regimes in which the convergence

---

[4]Code can be found at https://github.com/antonioalvarezl/2026-MLP-Attention-Energy.

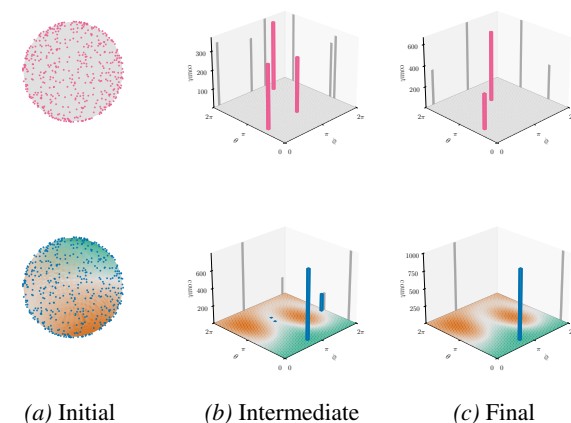

*(a)* Initial     *(b)* Intermediate     *(c)* Final

*Figure 4.* Gradient ascent on $\mathbb{S}^2$ with β = 1. **Top row:** pure self-attention. **Bottom row:** self-attention with ReLU perceptron. An animation is available at https://github.com/antonioalvarezl/2026-MLP-Attention-Energy/blob/main/examples/USAS2.gif

.

criteria are satisfied over long windows even though residual motion persists at larger timescales.

**Emergence of clusters**

We begin with gradient ascent on $\mathbb{S}^1$ with ReLU perceptron. Figure 3 displays three representative runs.

This protocol is repeated on $\mathbb{S}^2$. For visualization, we map each particle $x_i(t) \in \mathbb{S}^2$ to spherical angles $(\theta_i(t), \phi_i(t))$ and build a two-dimensional histogram on a uniform $(\theta, \phi)$–grid.

## 4.1. Gradient descent

We now turn to the minimization/descent dynamics, which is (2.6) with a − sign. Here pure self-attention is repulsive and promotes spreading of particles, while the perceptron enforces singular stationary configurations. As established in Proposition 3.7 **(ii)**, this system admits a unique global minimizer $\mu_*$ for every choice of parameters, and Theorem 3.3 applies to its stationary points.

We first run gradient descent with a ReLU perceptron: $\sigma(s) = s_+$. Figure 5 shows particle trajectories for $\beta \in \{0.05, 5, 50\}$ (recall that the convergence histogram for β = 1 was shown in Figure 1). A visible fraction of the mass remains trapped where the ReLU is identically zero: in those regions, the perceptron vanishes. Numerically, this produces a mixed stationary pattern: atomic clusters in active regions coexisting with frozen components in dead zones, consistent with Theorem 3.3**(i)**–**(iii)**.

We repeat the same experiment using GeLU. Figure 6 shows representative histograms at convergence.

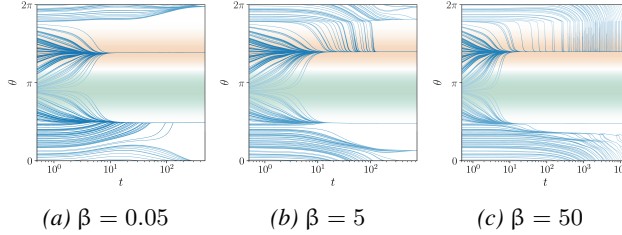

*Figure 5.* Gradient descent on $\mathbb{S}^1$ with ReLU perceptron.

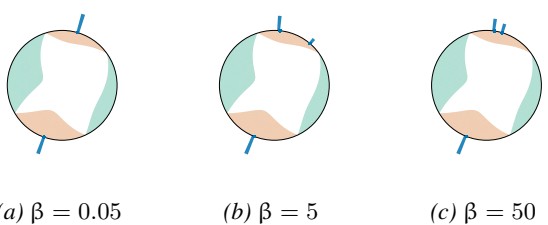

*(a)* β = 0.05     *(b)* β = 5     *(c)* β = 50

*Figure 6.* Histograms at final time for gradient descent with GeLU perceptron.

The analogous setup on $\mathbb{S}^2$ appears in Figure 7. Simulations for higher dimensions $d > 3$ are deferred to Appendix C.

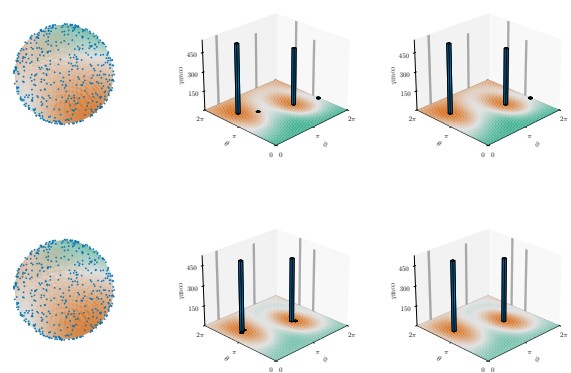

*Figure 7.* Gradient descent on $\mathbb{S}^2$ with β = 1. **Top row:** ReLU perceptron. **Bottom row:** GeLU perceptron. An animation is available at https://github.com/antonioalvarezl/2026-MLP-Attention-Energy/blob/main/examples/USAdS2.gif

.

### 4.2. Softmax normalization

We now study the dynamics governed by (2.9)—referred to as SA—across the settings previously considered on $\mathbb{S}^1$ and $\mathbb{S}^2$. Figure 8 illustrates the particle trajectories on $\mathbb{S}^1$ for three representative values of β.

### 4.3. Scaling of support size

We count the number of clusters (atoms) at convergence for every setup considered across the swept values of β. Results

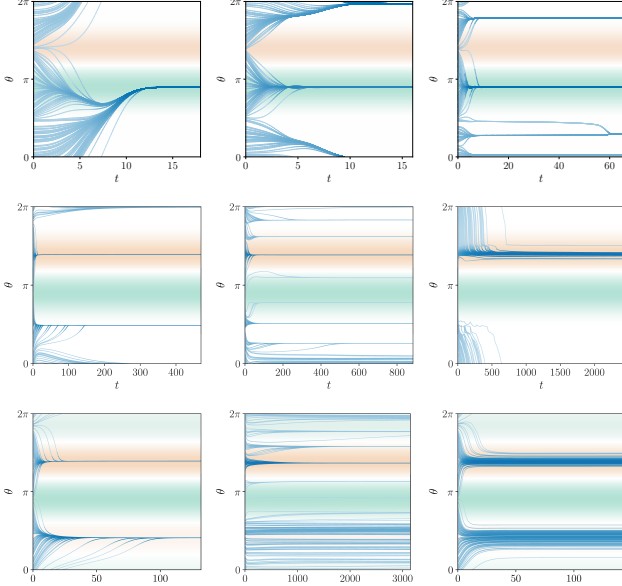

*Figure 8.* Trajectories following softmax-normalized attention. **Top row:** gradient ascent with ReLU perceptron. **Middle row:** gradient descent with ReLU perceptron. **Bottom row:** gradient descent with GeLU perceptron.

are summarized in Figure 9. For pure self-attention (setup S1-0), the reported cluster counts correspond to the number of atoms in the metastable configuration, rather than the long-time limit. Indeed, these dynamics are known to collapse the initial measure to a single Dirac mass (Geshkovski et al., 2025; Chen et al., 2025b; Geshkovski et al., 2024b).

The same caveat applies when adding a perceptron (setup S1): for the fixed ϑ used throughout, the perceptron potential exhibits a single local maximum (cf. Figure 3), so $\mathsf{E}_{β,ϑ}$ has a unique maximizer by Proposition 3.7. Yet, we still likely observe metastable configurations with more than one atom on computational time horizons.

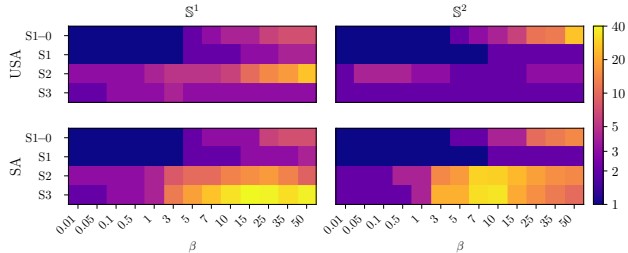

*Figure 9.* Number of clusters at convergence across setups. S1/S1-0: Ascent with/without ReLU perceptron. S2: Descent with ReLU perceptron. S3: Descent with GeLU perceptron.

### Sensitivity to initial configuration

We run 10 independent simulations at β = 10, using GeLU, with identical weights ϑ and resampled uniform initializa-

tions. Figure 10 compares three settings: (i) pure self-attention (no perceptron), (ii) gradient ascent with a perceptron, and (iii) descent with the same perceptron. In the absence of a perceptron, the cluster locations are highly seed-dependent. In contrast, the perceptron drift breaks this symmetry, anchoring the clusters to specific locations that are largely independent of the initial configuration.

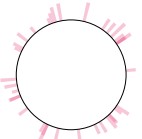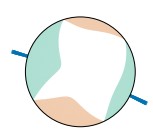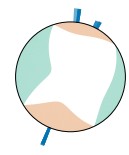

*Figure 10.* Superposition of stationary measures from 10 independent uniform initializations with $\beta = 10$ and $\sigma = $ GeLU. **Left:** pure self-attention descent yields seed-dependent cluster locations. **Middle:** the perceptron drift in ascent breaks the symmetry and selects seed-independent clusters. **Right:** in descent, the perceptron still anchors a seed-independent stationary support, which is less concentrated and may include spurious atoms.

## Acknowledgements

This project was initiated during a research stay of A.Á. visiting B.G. at Laboratoire Jacques-Louis Lions in Paris. A.Á. acknowledges financial support from the Spanish National Research Council (CSIC) through the iMOVE 2024 mobility programme (reference IMOVE24243), and has also been funded by the project PID2023-146872OB-I00 of MICIU (Spain).

## Impact Statement

This paper presents work whose goal is to advance the field of Machine Learning. There are many potential societal consequences of our work, none of which we feel must be specifically highlighted here.

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

# A. Proofs

## A.1. Preliminaries

We denote the spherical harmonics in $L^2(\sigma_d)$ by $\{Y_{j\ell}\}_{\substack{j \geqslant 0 \\ 1 \leqslant \ell \leqslant N_j}}$ where the dimension $N_j$ is given by

$$N_j := \binom{d+j-1}{j} - \binom{d+j-3}{j-2}.$$

Under the standard convention that $\binom{n}{k} = 0$ for $k < 0$, we have $N_0 = 1$ and $N_1 = d$.

By the Funk–Hecke identity, the spherical harmonics are precisely the eigenfunctions:

$$\int e^{\beta x \cdot y} Y_{j\ell}(x) \, d\sigma_d(x) = \lambda_j(\beta) \, Y_{j\ell}(y). \tag{A.1}$$

For $\beta > 0$, the eigenvalues $\lambda_j(\beta)$ are strictly positive and explicitly given by

$$\lambda_j(\beta) := \Gamma\left(\frac{d}{2}\right)\left(\frac{2}{\beta}\right)^{\frac{d}{2}-1} I_{j+\frac{d}{2}-1}(\beta), \qquad j \geqslant 0,$$

where $I_\alpha$ is the modified Bessel function of the first kind of order $\alpha$.

**Lemma A.1.** *For any $\beta > 0$, the map $\mu \in \mathcal{P}(\mathbb{S}^{d-1}) \mapsto f^\mu \in C^\infty(\mathbb{S}^{d-1})$, defined by*

$$f^\mu(x) := \int e^{\beta x \cdot y} \, d\mu(y),$$

*is injective. Moreover, if $f^\mu = \sum_{j=0}^\infty \sum_{\ell \in [\![1,N_j]\!]} \widehat{f^\mu_{j\ell}} Y_{j\ell}$ is its spherical harmonic expansion, the following properties hold:*

*(i) $\mu$ is absolutely continuous with respect to $\sigma_d$ with density $\frac{d\mu}{d\sigma_d} \in L^2(\sigma_d)$ if and only if*

$$\sum_{j=0}^\infty \sum_{\ell \in [\![1,N_j]\!]} \left| \frac{\widehat{f^\mu_{j\ell}}}{\lambda_j(\beta)} \right|^2 < \infty.$$

*In this case, the density is given by the $L^2(\sigma_d)$-convergent series*

$$\frac{d\mu}{d\sigma_d}(x) = \sum_{j=0}^\infty \sum_{\ell \in [\![1,N_j]\!]} \frac{\widehat{f^\mu_{j\ell}}}{\lambda_j(\beta)} Y_{j\ell}(x). \tag{A.2}$$

*In particular, $f^\mu$ is a polynomial of degree $k$ if and only if $\mu$ has a density that is a polynomial of degree $k$.*

*(ii) $f^\mu$ is an even function if and only if $\mu(A) = \mu(-A)$ for every Borel set $A \subset \mathbb{S}^{d-1}$.*

*Proof.* We divide the proof into three parts.

**Injectivity.** Let $\mu_1, \mu_2 \in \mathcal{P}(\mathbb{S}^{d-1})$ and assume

$$\int e^{\beta x \cdot y} \, d\mu_1(y) = \int e^{\beta x \cdot y} \, d\mu_2(y) \quad \text{for all } x \in \mathbb{S}^{d-1}.$$

Set $\nu := \mu_1 - \mu_2$ and define $f^\nu(x) := \int e^{\beta x \cdot y} \, d\nu(y)$; then $f^\nu \equiv 0$. For each $(j, \ell)$, using Fubini's theorem and (A.1), we compute the Fourier coefficients of $f^\nu$:

$$0 = \widehat{f^\nu_{j\ell}} = \int f^\nu(x) Y_{j\ell}(x) \, d\sigma_d(x) = \int \left( \int e^{\beta x \cdot y} Y_{j\ell}(x) \, d\sigma_d(x) \right) d\nu(y)$$

$$= \lambda_j(\beta) \int Y_{j\ell}(y) \, d\nu(y). \tag{A.3}$$

Since $\lambda_j(\beta) > 0$, it follows that $m_{j\ell}(\nu) := \int Y_{j\ell} \, d\nu = 0$ for all $j, \ell$. Hence, for any finite spherical polynomial $P = \sum a_{j\ell} Y_{j\ell}$, we have

$$\int P \, d\nu = \sum a_{j\ell} \, m_{j\ell}(\nu) = 0.$$

By the Stone–Weierstrass theorem, spherical polynomials are uniformly dense in $C^0(\mathbb{S}^{d-1})$. Since $\nu$ is a finite signed measure, uniform approximation implies $\int h \, d\nu = 0$ for all $h \in C^0(\mathbb{S}^{d-1})$. By the uniqueness of the Riesz representation theorem we deduce that $\nu = 0$, and thus $\mu_1 = \mu_2$.

For later use, note that applying the computation of (A.3) to $\mu$ instead of $\nu$ yields the relation

$$\widehat{f}_{j\ell}^{\mu} = \lambda_j(\beta) \, m_{j\ell}, \qquad \text{where} \quad m_{j\ell} := \int Y_{j\ell} \, d\mu. \tag{A.4}$$

**Proof of (i).** Assume $\sum_{j,\ell} \left| \widehat{f}_{j\ell}^{\mu} / \lambda_j(\beta) \right|^2 < \infty$. Since $\{Y_{j\ell}\}_{j,\ell}$ is an orthonormal basis of $L^2(\sigma_d)$, there exists a unique function $g \in L^2(\sigma_d)$ such that

$$g(x) := \sum_{j=0}^{\infty} \sum_{\ell \in [\![1, N_j]\!]} \frac{\widehat{f}_{j\ell}^{\mu}}{\lambda_j(\beta)} Y_{j\ell}(x)$$

with convergence in $L^2(\sigma_d)$. In particular, $g \in L^1(\sigma_d)$, so $g \, \sigma_d$ defines a finite signed measure on $\mathbb{S}^{d-1}$. For any finite spherical polynomial $P = \sum a_{j\ell} Y_{j\ell}$, using (A.4),

$$\int P \, d\mu = \sum a_{j\ell} m_{j\ell} = \sum a_{j\ell} \frac{\widehat{f}_{j\ell}^{\mu}}{\lambda_j(\beta)} = \int P \, g \, d\sigma_d.$$

Since spherical polynomials are dense in $C^0(\mathbb{S}^{d-1})$, this identity extends to all $h \in C^0(\mathbb{S}^{d-1})$. By the uniqueness of the Riesz representation theorem, $d\mu = g \, d\sigma_d$, which yields (A.2). Conversely, if $\mu$ has a density $\rho \in L^2(\sigma_d)$ so that $d\mu = \rho \, d\sigma_d$, then $m_{j\ell} = \langle \rho, Y_{j\ell} \rangle_{L^2}$. Parseval's identity then gives

$$\sum_{j,\ell} \left| \frac{\widehat{f}_{j\ell}^{\mu}}{\lambda_j(\beta)} \right|^2 = \sum_{j,\ell} |m_{j\ell}|^2 = \|\rho\|_{L^2(\sigma_d)}^2 < \infty.$$

Finally, $f^{\mu}$ is a spherical polynomial of degree $k$ if and only if $\widehat{f}_{j\ell}^{\mu} = 0$ for $j > k$ and $\widehat{f}_{k\ell}^{\mu} \neq 0$, which, by (A.2), is equivalent to the density being a spherical polynomial of degree $k$.

**Proof of (ii).** If $\mu(A) = \mu(-A)$ for every Borel set $A \subset \mathbb{S}^{d-1}$, then for every $x \in \mathbb{S}^{d-1}$,

$$f^{\mu}(-x) = \int e^{\beta(-x)\cdot y} \, d\mu(y) = \int e^{\beta x \cdot (-y)} \, d\mu(y) = \int e^{\beta x \cdot y} \, d\mu(y) = f^{\mu}(x),$$

so $f^{\mu}$ is even. Conversely, assume $f^{\mu}(-x) = f^{\mu}(x)$ for all $x \in \mathbb{S}^{d-1}$. Define $\mu^- := (-\mathrm{Id})_{\#}\mu$. Then, for all $x \in \mathbb{S}^{d-1}$,

$$\int e^{\beta x \cdot y} \, d\mu^-(y) = \int e^{\beta x \cdot (-y)} \, d\mu(y) = \int e^{\beta(-x)\cdot y} \, d\mu(y)$$

$$= f^{\mu}(-x) = f^{\mu}(x) = \int e^{\beta x \cdot y} \, d\mu(y).$$

By the injectivity proved above, we conclude that $\mu^- = \mu$, meaning $\mu(A) = \mu(-A)$ for every Borel set $A \subset \mathbb{S}^{d-1}$. $\qquad \square$

*Remark* A.2. Fix a symmetric nonsingular matrix $B \in \mathbb{R}^{d \times d}$ and define, for $\mu \in \mathcal{P}(\mathbb{S}^{d-1})$,

$$f_B(x) := \int e^{x^\top B y} \, d\mu(y), \qquad x \in \mathbb{S}^{d-1}.$$

Then the statements of injectivity and the parity characterization **(ii)** in Lemma A.1 remain valid for $f_B$, with essentially the same proof and only the following modifications.

**Injectivity.** Let $\mu_1, \mu_2 \in \mathcal{P}(\mathbb{S}^{d-1})$ and assume $\int e^{x^\top By} \, d\mu_1(y) = \int e^{x^\top By} \, d\mu_2(y)$ for all $x \in \mathbb{S}^{d-1}$. Set $\nu := \mu_1 - \mu_2$ and define

$$F_\nu(z) := \int e^{z \cdot y} \, d\nu(y), \qquad z \in \mathbb{R}^d.$$

Then $F_\nu \equiv 0$ on the boundary of the ellipsoid $\Omega := \{B^\top x \ : \ \|x\| < 1\}$. Moreover, differentiating under the integral sign shows that $(\Delta - 1)F_\nu = 0$ on $\mathbb{R}^d$, since for any fixed $y \in \mathbb{S}^{d-1}$, the map $z \mapsto e^{z \cdot y}$ satisfies

$$\Delta(e^{z \cdot y}) = \|y\|^2 e^{z \cdot y} = e^{z \cdot y}.$$

By uniqueness for the Dirichlet problem for $\Delta - 1$ on $\Omega$, we obtain $F_\nu \equiv 0$ in $\Omega$. Hence $F_\nu$ vanishes in a neighborhood of $0$, and therefore on all of $\mathbb{R}^d$ by real-analyticity. Expanding at $0$, we get $\int P(y) \, d\nu(y) = 0$ for every polynomial $P$. Since the restrictions of polynomials to $\mathbb{S}^{d-1}$ are uniformly dense in $C^0(\mathbb{S}^{d-1})$, it follows that $\nu = 0$, and thus $\mu_1 = \mu_2$.

**Parity.** The proof that $f_B^\mu$ is even if and only if $\mu(A) = \mu(-A)$ for every Borel set $A \subset \mathbb{S}^{d-1}$ is exactly the same as in Lemma A.1(**ii**), using the injectivity of $f_B^\mu$ established above.

We emphasize that (A.4) and the $L^2$ inversion criterion in Lemma A.1(**i**) rely on the isotropic (zonal) kernel $e^{\beta x \cdot y}$, failing as soon as $B$ introduces anisotropy.

**Lemma A.3.** *Fix $d = 2$ and $\beta > 0$. Let $\mathsf{K}_\beta(\theta) := e^{\beta \cos \theta}$ for $\theta \in (-\pi, \pi]$, and define*

$$\theta_c(\beta) := \arccos\left(\frac{\sqrt{1 + 4\beta^2} - 1}{2\beta}\right) \in (-\pi, \pi]. \tag{A.5}$$

*Then $\mathsf{K}_\beta$ is strictly concave on $(-\theta_c(\beta), \theta_c(\beta))$, and*

$$\theta_c(\beta) = \beta^{-1/2} + O(\beta^{-3/2}) \quad \text{as } \beta \to \infty, \qquad \theta_c(\beta) \to \frac{\pi}{2} \quad \text{as } \beta \to 0^+.$$

*Moreover, for each $\lambda \in (0, 1)$ there exists $\beta_0 = \beta_0(\lambda) > 0$ such that*

$$\sup_{|\theta| \leqslant \lambda \theta_c(\beta)} \mathsf{K}_\beta''(\theta) \leqslant -e^{-\lambda^2/2} \frac{1 - \lambda^2}{2} \beta e^\beta \qquad \text{for all } \beta \geqslant \beta_0(\lambda), \tag{A.6}$$

*and there exists $\beta_1 > 0$ such that*

$$\max_{\theta \in [\theta_c(\beta), \pi]} \mathsf{K}_\beta''(\theta) \leqslant 2\beta e^{\beta - 3/2} \qquad \text{for all } \beta \geqslant \beta_1. \tag{A.7}$$

*Proof.* A direct computation gives
$$\mathsf{K}_\beta''(\theta) = e^{\beta \cos \theta}(\beta^2 \sin^2 \theta - \beta \cos \theta). \tag{A.8}$$

Thus $\mathsf{K}_\beta''(\theta) = 0$ is equivalent to $\beta \cos^2 \theta + \cos \theta - \beta = 0$, whose positive root is precisely

$$\cos \theta_c(\beta) = \frac{\sqrt{1 + 4\beta^2} - 1}{2\beta}, \tag{A.9}$$

so $\theta_c(\beta) \in (0, \pi/2)$ is well defined (the other root being $< -1$). Because $\mathsf{K}_\beta''(0) = -\beta e^\beta < 0$ and $\theta_c$ is the first positive root, $\mathsf{K}_\beta$ is strictly concave on $(-\theta_c(\beta), \theta_c(\beta))$.

From (A.9), we obtain
$$\frac{\sqrt{1 + 4\beta^2} - 1}{2\beta} = 1 - \frac{1}{2\beta} + O(\beta^{-2}) \qquad \text{as } \beta \to \infty,$$

yielding $\theta_c(\beta) = \beta^{-1/2} + O(\beta^{-3/2})$ via the expansion $\cos \theta = 1 - \theta^2/2 + O(\theta^4)$. Similarly,

$$\frac{\sqrt{1 + 4\beta^2} - 1}{2\beta} = \beta + O(\beta^3) \qquad \text{as } \beta \to 0^+,$$

hence $\cos\theta_c(\beta) \to 0$ and $\theta_c(\beta) \to \pi/2$.

To prove (A.6), we show that $\mathsf{K}''_\beta$ is strictly increasing on $[0, \theta_c(\beta)]$. Differentiating (A.8) gives

$$\mathsf{K}'''_\beta(\theta) = -\beta\sin\theta\,\mathsf{K}''_\beta(\theta) + e^{\beta\cos\theta}\beta\sin\theta\,(2\beta\cos\theta + 1). \tag{A.10}$$

For $0 < \theta < \theta_c(\beta)$, we have $\sin\theta > 0$, $\cos\theta > 0$, and $\mathsf{K}''_\beta(\theta) < 0$. Thus both terms on the right-hand side are positive, yielding $\mathsf{K}'''_\beta(\theta) > 0$. Since $\mathsf{K}''_\beta$ is even, its supremum on $[-\lambda\theta_c, \lambda\theta_c]$ for every $\lambda \in (0,1)$ is attained at the boundaries. Using the expansion of $\theta_c(\beta)$, we have

$$\cos(\lambda\theta_c) = 1 - \frac{\lambda^2}{2\beta} + O(\beta^{-2}) \qquad \text{and} \qquad \sin^2(\lambda\theta_c) = \frac{\lambda^2}{\beta} + O(\beta^{-2}).$$

Substituting into (A.8) yields

$$\begin{aligned}
\mathsf{K}''_\beta(\lambda\theta_c) &= e^{\beta\cos(\lambda\theta_c)}\left(\beta^2\sin^2(\lambda\theta_c) - \beta\cos(\lambda\theta_c)\right) \\
&= e^{\beta - \lambda^2/2}\left(1 + O(\beta^{-1})\right)\left(\beta(\lambda^2 - 1) + \frac{\lambda^2}{2} + O(1)\right) \\
&= -\left(1 - \lambda^2\right)e^{\beta - \lambda^2/2}\beta\left(1 + O(\beta^{-1})\right).
\end{aligned}$$

Therefore, for $\beta$ sufficiently large, we obtain the bound in (A.6).

To prove (A.7), we find the maximum of $\mathsf{K}''_\beta$ on $[\theta_c, \pi]$. Rearranging (A.10), the condition $\mathsf{K}'''_\beta(\theta) = 0$ on $(0, \pi)$ reduces to $q_\beta(\cos\theta) = 0$, where $q_\beta(t) := \beta^2 t^2 + 3\beta t - \beta^2 + 1$. The unique root in $(-1, \cos\theta_c(\beta))$ is

$$t_* = \frac{-3 + \sqrt{4\beta^2 + 5}}{2\beta}.$$

Let $\theta_* \in (\theta_c, \pi)$ be defined by $\cos\theta_* = t_*$. Since $\mathsf{K}'''_\beta(\theta)$ changes sign from positive to negative at $\theta_*$, the maximum is attained there. Because $q_\beta(t_*) = 0$, we can express the quadratic term as $\beta^2(1 - t_*^2) = 3\beta t_* + 1$. Substituting this into (A.8) simplifies the evaluation to

$$\mathsf{K}''_\beta(\theta_*) = e^{\beta t_*}(2\beta t_* + 1).$$

Finally, expanding $t_* = 1 - \frac{3}{2\beta} + \frac{5}{8\beta^2} + O(\beta^{-4})$ for large $\beta$ provides

$$\begin{aligned}
\mathsf{K}''_\beta(\theta_*) &= e^{\beta - \frac{3}{2}}\left(1 + \frac{5}{8\beta} + O(\beta^{-2})\right)2\beta\left(1 - \frac{1}{\beta} + O(\beta^{-2})\right) \\
&= 2\beta e^{\beta - \frac{3}{2}}\left(1 - \frac{3}{8\beta} + O(\beta^{-2})\right).
\end{aligned}$$

Since $1 - \frac{3}{8\beta} < 1$ for large $\beta$, this implies (A.7) for all $\beta \geqslant \beta_1$. $\qquad\square$

## A.2. Proof of Theorem 3.1 and Theorem 3.2

*Proof of Theorem 3.1.* Set $x(\theta) = (\cos\theta, \sin\theta)$ and $\tau(\theta) = (-\sin\theta, \cos\theta)$. For $\theta \in [0, 2\pi)$ define

$$\begin{aligned}
f_\beta(\theta) &:= \int e^{\beta\cos(\phi - \theta)}\sin(\phi - \theta)\,\mathrm{d}\mu(\phi), \\
f_\vartheta(\theta) &:= \sum_{j \in [\![1,2]\!]} \omega_j\,(a_j \cdot x(\theta))_+\,a_j \cdot \tau(\theta),
\end{aligned}$$

and set $g := f_\beta + f_\vartheta$, corresponding to the gradient $\nabla\frac{\delta\mathsf{E}_\beta}{\delta\mu}[\mu] + u_\vartheta$ expressed in angular coordinates.

Let $\mathscr{Z} := \{\theta \in [0, 2\pi) : a_j \cdot x(\theta) = 0 \text{ for some } j\}$. Each equation $a_j \cdot x(\theta) = 0$ has at most two solutions, so $\mathscr{Z}$ is finite. On any connected component $I \subset [0, 2\pi) \setminus \mathscr{Z}$, the active set

$$J_I := \{j : a_j \cdot x(\theta) > 0 \text{ for all } \theta \in I\}$$

is constant, and for $\theta \in I$ we have

$$(a_j \cdot x(\theta))_+ = \mathbf{1}_{j \in J_I}\, a_j \cdot x(\theta), \qquad \frac{\mathrm{d}}{\mathrm{d}\theta}\, (a_j \cdot x(\theta))_+ = \mathbf{1}_{j \in J_I}\, a_j \cdot \tau(\theta).$$

Thus $f_\vartheta$ is real–analytic on each $I$. On the other hand, since

$$\mathsf{K}_\beta\,(\psi) := e^{\beta \cos \psi} = I_0(\beta) + 2 \sum_{n \geqslant 1} I_n(\beta) \cos(n\psi),$$

its Fourier series is absolutely convergent. Hence the convolution with the measure $\mu$,

$$(\mathsf{K}_\beta * \mu)\,(\theta) := \int \mathsf{K}_\beta\,(\theta - \phi)\,\mathrm{d}\mu(\phi),$$

is real-analytic on $\mathbb{S}^1$, and $f_\beta(\theta) = \frac{1}{\beta} \frac{\mathrm{d}}{\mathrm{d}\theta}(\mathsf{K}_\beta * \mu)(\theta)$ is real-analytic as well. All in all, we deduce that $g$ is continuous on $[0, 2\pi)$ and real-analytic on each $I$.

For each component $I$, from (2.8) we get

$$\operatorname{supp}\mu \cap I \subset \{\theta \in I : g(\theta) = 0\}. \tag{A.11}$$

Assume, for a contradiction, that $\operatorname{supp}\mu$ is infinite. Then $\operatorname{supp}\mu \cap I$ must be infinite for at least one of the finitely many components $I$. Define the global real-analytic function

$$\widetilde{g}\,(\theta) := f_\beta\,(\theta) + \sum_{j \in J_I} \omega_j \,\langle a_j, x(\theta)\rangle \,\langle a_j, \tau(\theta)\rangle,$$

which satisfies $\widetilde{g}(\theta) = g(\theta)$ for all $\theta \in I$. Hence, by (A.11), the zero set of $\widetilde{g}$ contains infinitely many points in $I$ and thus has an accumulation point in $\mathbb{S}^1$. By the identity theorem for real-analytic functions, it follows that $\widetilde{g} \equiv 0$ on $\mathbb{S}^1$. Equivalently,

$$f_\beta\,(\theta) = - \sum_{j \in J_I} \omega_j \,\langle a_j, x(\theta)\rangle \,\langle a_j, \tau(\theta)\rangle \qquad \text{for all } \theta \in \mathbb{S}^1.$$

Integrating in $\theta$, and recalling $f_\beta = \frac{1}{\beta}(\mathsf{K}_\beta * \mu)'$, we find that $g = H'$ on $I$, for

$$H\,(\theta) := \frac{1}{\beta}\,(\mathsf{K}_\beta * \mu)\,(\theta) + \sum_j \frac{\omega_j}{2}\,(a_j \cdot x(\theta))_+^2.$$

Since $\widetilde{g} \equiv 0$ on $\mathbb{S}^1$, we obtain the global identity

$$(\mathsf{K}_\beta * \mu)\,(\theta) = C_I - \beta \sum_{j \in J_I} \frac{\omega_j}{2}\,(a_j \cdot x(\theta))^2 \qquad (\text{for } \theta \in [0, 2\pi)) \tag{A.12}$$

for some constant $C_I$.

By Lemma A.1(i), (A.12) implies that $\mu$ is absolutely continuous with an $L^2$-density which is a trigonometric polynomial of degree $\leqslant 2$. As a nonnegative real-analytic function on $\mathbb{S}^1$, this density cannot vanish on an open arc unless it is identically zero; hence $\operatorname{supp}\mu = \mathbb{S}^1$.

Let $I'$ be a component adjacent to $I$, and let $\theta_*$ be the common boundary point. Since $\operatorname{supp}\mu = \mathbb{S}^1$, we have $\operatorname{supp}\mu \cap I'$ is infinite, and repeating the argument used on $I$ gives another global identity

$$(\mathsf{K}_\beta * \mu)\,(\theta) = C_{I'} - \beta \sum_{j \in J_{I'}} \frac{\omega_j}{2}\,(a_j \cdot x(\theta))^2 \qquad (\text{for } \theta \in [0, 2\pi))\,. \tag{A.13}$$

Let $B := J_I \triangle J_{I'}$ denote the symmetric difference. Clearly $B \neq \varnothing$, since $J_I \neq J_{I'}$ because at least one index changes sign when crossing $\theta_*$. Subtracting (A.13) from (A.12) we obtain

$$\frac{\beta}{2} \sum_{j \in B} \xi_j\, \omega_j\,(a_j \cdot x(\theta))^2 \equiv C_I - C_{I'} \qquad (\text{for } \theta \in [0, 2\pi))\,, \tag{A.14}$$

where $\xi_j \in \{\pm 1\}$ records whether $j$ is added or removed when passing from $I$ to $I'$. For every $j \in B$ we have $a_j \cdot x(\theta_*) = 0$, hence evaluating (A.14) at $\theta = \theta_*$ yields $C_I = C_{I'}$ and therefore

$$\sum_{j \in B} \xi_j \, \omega_j \, (a_j \cdot x(\theta))^2 \; \equiv \; 0 \qquad (\text{for } \theta \in [0, 2\pi)). \tag{A.15}$$

The left-hand side of (A.15) is exactly the difference of $(v_\vartheta \circ x)(\theta)$ on $I$ and $I'$, which therefore vanishes identically on $[0, 2\pi)$. Hence $(v_\vartheta \circ x)(\theta)$ coincides on $\overline{I \cup I'}$ with a single trigonometric polynomial of degree at most 2.

Since this argument applies to every adjacent pair of components, it follows by connectivity that $v_\vartheta \circ x$ is given in the full $\mathbb{S}^1$ by a single trigonometric polynomial of degree at most 2. In particular, $v_\vartheta$ is real-analytic on $\mathbb{S}^1$, contradicting our assumption. This shows that $\mathrm{supp}\,\mu$ is finite, and combining this with (A.11) yields that $\mu$ is purely atomic with finitely many atoms.

$\square$

*Remark* A.4. The conclusion of Theorem 3.1 holds for any $\sigma$ globally Lipschitz, piecewise polynomial (e.g., the leaky ReLU), provided $v_\vartheta$ is not real-analytic.

The proof applies *mutatis mutandis*. On each component $I$, the field $f_\vartheta$ now becomes a trigonometric polynomial of some finite degree. Consequently, if the support accumulates in $I$, analytic continuation implies that the global identity (A.12) holds with a polynomial of some finite degree. Lemma A.1**(i)** then guarantees a global polynomial density, forcing $\mathrm{supp}\,\mu = \mathbb{S}^1$. The subtraction argument via (A.14) yields the contradiction exactly as before.

The same extension applies to Theorem 3.3.

*Proof of Theorem 3.2.* Let $\mu \in \mathcal{P}(\mathbb{S}^1)$ be a strict SOPD Wasserstein critical point of $\mathsf{E}_{\beta,\vartheta}$. In particular, $\mathrm{Hess}_\mu \mathsf{E}_{\beta,\vartheta}$ is well-defined at $\mu$. Write $x(\theta) = (\cos\theta, \sin\theta)$ and $\mathsf{K}_\beta(\phi) = e^{\beta \cos \phi}$, and define

$$H(\theta) := \frac{1}{\beta} (\mathsf{K}_\beta * \mu)(\theta) + \frac{1}{2} (v_\vartheta \circ x)(\theta).$$

Since $\sigma$ is real-analytic, the potential $\theta \mapsto (v_\vartheta \circ x)(\theta)$ is real-analytic on $\mathbb{S}^1$. The convolution term $(\mathsf{K}_\beta * \mu)$ is also real-analytic. Hence $H$ is real-analytic on $\mathbb{S}^1$.

Assume for contradiction that $\mathrm{supp}\,\mu$ is infinite. Then $\mathrm{supp}\,\mu$ has an accumulation point in $\mathbb{S}^1$. The first-order condition (2.8) forces $H'$ to vanish on $\mathrm{supp}\,\mu$. Since $H'$ is real-analytic and vanishes on a set with an accumulation point, the identity theorem implies $H' \equiv 0$ on $\mathbb{S}^1$. Consequently, $H'' \equiv 0$ on $\mathbb{S}^1$, which gives the global cancellation:

$$\frac{1}{\beta} (\mathsf{K}_\beta'' * \mu)(\theta) + \frac{1}{2} \partial_\theta^2 (v_\vartheta \circ x)(\theta) \; = \; 0 \qquad (\text{for } \theta \in \mathbb{S}^1). \tag{A.16}$$

To compute the second variation, let $\xi \in \mathsf{T}_\mu \mathcal{P}(\mathbb{S}^1)$. Since the Wasserstein Hessian is well-defined at $\mu$,

$$\mathrm{Hess}_\mu \mathsf{E}_{\beta,\vartheta}(\xi, \xi) = \frac{1}{2\beta} \iint \mathsf{K}_\beta''(\theta - \phi)(\xi(\theta) - \xi(\phi))^2 \, \mathrm{d}\mu(\theta) \, \mathrm{d}\mu(\phi)$$
$$+ \frac{1}{2} \int \partial_\theta^2 (v_\vartheta \circ x)(\theta) \, \xi(\theta)^2 \, \mathrm{d}\mu(\theta). \tag{A.17}$$

Expanding the square using the symmetry of $\mathsf{K}_\beta''$, we rewrite this as:

$$\mathrm{Hess}_\mu \mathsf{E}_{\beta,\vartheta}(\xi, \xi) = \int \left[ \frac{1}{\beta} (\mathsf{K}_\beta'' * \mu)(\theta) + \frac{1}{2} \partial_\theta^2 (v_\vartheta \circ x)(\theta) \right] \xi(\theta)^2 \, \mathrm{d}\mu(\theta)$$
$$- \frac{1}{\beta} \iint \mathsf{K}_\beta''(\theta - \phi) \, \xi(\theta)\xi(\phi) \, \mathrm{d}\mu(\theta) \, \mathrm{d}\mu(\phi). \tag{A.18}$$

By (A.16), the term in brackets vanishes identically. Thus, the Hessian reduces to a bilinear form:

$$\mathrm{Hess}_\mu \mathsf{E}_{\beta,\vartheta}(\xi, \xi) = -\frac{1}{\beta} \iint \mathsf{K}_\beta''(\theta - \phi) \, \xi(\theta)\xi(\phi) \, \mathrm{d}\mu(\theta) \, \mathrm{d}\mu(\phi). \tag{A.19}$$

We construct a sequence of tangent vectors along which (A.19) is arbitrarily small, thereby contradicting strict positivity.

Fix a small arc $J_\delta \subset \mathbb{S}^1$ centered at an accumulation point of $\operatorname{supp} \mu$, with diameter $\leqslant \delta$, so that $\mu(J_\delta) > 0$. Pick two disjoint subarcs $I_1, I_2 \subset J_\delta$ with $\mu(I_i) > 0$. Choose $u_i \in C_c^\infty(I_i)$ non-constant, view it as a smooth function on $\mathbb{S}^1$ by extending it by zero outside $I_i$, and set $\eta_i := u_i'$. Then $\eta_i \in \mathsf{T}_\mu \mathcal{P}(\mathbb{S}^1)$ (recall (2.3)) and $\eta_i$ is supported in $I_i$. Moreover, by choosing $u_i$ so that $u_i'$ does not vanish on a neighborhood of some point in $\operatorname{supp} \mu \cap I_i$, we ensure $\|\eta_i\|_{L^2(\mu)} > 0$.

Set $m_i := \int \eta_i \, d\mu$. If $(m_1, m_2) \neq (0, 0)$, define

$$\xi_0 := m_2 \, \eta_1 - m_1 \, \eta_2,$$

so that $\int \xi_0 \, d\mu = m_2 m_1 - m_1 m_2 = 0$. If $m_1 = m_2 = 0$, simply set $\xi_0 := \eta_1$ (which already satisfies $\int \xi_0 \, d\mu = 0$). In either case, since $\eta_1$ and $\eta_2$ have disjoint supports and $\xi_0$ is not identically zero, we have $\xi_0 \not\equiv 0$.

Normalize $\xi_\delta := \xi_0 / \|\xi_0\|_{L^2(\mu)}$. Then

$$\xi_\delta \in \mathsf{T}_\mu \mathcal{P}(\mathbb{S}^1), \qquad \operatorname{supp} \xi_\delta \subset J_\delta, \qquad \text{and} \qquad \|\xi_\delta\|_{L^2(\mu)} = 1.$$

Moreover, by construction, $\int \xi_\delta \, d\mu = 0$. Evaluating (A.19) at $\xi = \xi_\delta$:

$$\operatorname{Hess}_\mu \mathsf{E}_{\beta, \vartheta}(\xi_\delta, \xi_\delta) = -\frac{1}{\beta} \iint_{J_\delta \times J_\delta} \mathsf{K}_\beta''(\theta - \phi) \, \xi_\delta(\theta) \xi_\delta(\phi) \, d\mu(\theta) \, d\mu(\phi).$$

Since $\int \xi_\delta \, d\mu = 0$, adding a constant to the kernel does not change the integral. We replace $\mathsf{K}_\beta''(\theta - \phi)$ by $\mathsf{K}_\beta''(\theta - \phi) - \mathsf{K}_\beta''(0)$. Let $\omega(\delta) := \sup_{|h| \leqslant \delta} |\mathsf{K}_\beta''(h) - \mathsf{K}_\beta''(0)|$ for $\delta \in [0, \pi)$. Then,

$$|\operatorname{Hess}_\mu \mathsf{E}_{\beta, \vartheta}(\xi_\delta, \xi_\delta)| \leqslant \frac{1}{\beta} \iint |\mathsf{K}_\beta''(\theta - \phi) - \mathsf{K}_\beta''(0)| \, |\xi_\delta(\theta)| \, |\xi_\delta(\phi)| \, d\mu(\theta) \, d\mu(\phi)$$

$$\leqslant \frac{\omega(\delta)}{\beta} \|\xi_\delta\|_{L^1(\mu)}^2.$$

Using Cauchy-Schwarz, $\|\xi_\delta\|_{L^1(\mu)}^2 \leqslant \|\xi_\delta\|_{L^2(\mu)}^2 = 1$. Thus, the Hessian is bounded by $\omega(\delta)/\beta$, which tends to 0 as $\delta \to 0$. This implies

$$\inf \left\{ \operatorname{Hess}_\mu \mathsf{E}_{\beta, \vartheta}(\xi, \xi) \, : \, \xi \in \mathsf{T}_\mu \mathcal{P}(\mathbb{S}^1), \, \|\xi\|_{L^2(\mu)} = 1 \right\} = 0,$$

contradicting (2.5). Therefore $\operatorname{supp} \mu$ must be finite, and $\mu$ is purely atomic with finitely many atoms.

$\square$

## A.3. Proof of Theorem 3.3

*Proof of Theorem 3.3.* We split the proof into two parts.

PART I. NON-ABSOLUTE CONTINUITY

We reuse the definitions of $(\mathscr{Z}, I, J_I)$ introduced in (3.5)–(3.6).

Let $\mu \in \mathcal{P}(\mathbb{S}^{d-1})$ solve (2.8). Assume first that $\sigma(s) = s_+$ and $\mathsf{v}_\vartheta$ is not real-analytic on $\mathbb{S}^{d-1}$. On $I$ the signs of $x \mapsto a_j \cdot x$ are fixed, hence

$$\mathsf{v}_\vartheta(x) = \sum_{j \in [\![1, d]\!]} \omega_j \, (a_j \cdot x)_+^2 = \sum_{j \in J_I} \omega_j \, (a_j \cdot x)^2 \qquad \text{(for } x \in I). \tag{A.20}$$

By using the Weierstrass $M$-test on the expansion

$$e^{\beta x \cdot y} = \sum_{m \geqslant 0} \frac{\beta^m}{m!} (x \cdot y)^m,$$

we deduce that $\frac{\delta \mathsf{E}_\beta}{\delta \mu}[\mu]$ is real-analytic on $\mathbb{S}^{d-1}$. Thus, $H := \frac{\delta \mathsf{E}_{\beta, \vartheta}}{\delta \mu}[\mu]$ and $g := \nabla H$ are both real-analytic on every $I$.

For each $I$, from (2.8) we have

$$\operatorname{supp} \mu \cap I \subset \{x \in I : g(x) = 0\}. \tag{A.21}$$

Assume for contradiction that $\sigma_d(\operatorname{supp}\mu) > 0$. Since $\sigma_d(\mathscr{Z}) = 0$, we deduce that $\sigma_d(\operatorname{supp}\mu \cap I) > 0$ for some component $I$. Then by (A.21), the zero set of $g$ has positive measure in $I$, hence $g \equiv 0$ on $I$ by analyticity (Mityagin, 2020). Thus $H$ is constant on $I$ and, using (A.20),

$$\frac{\delta E_\beta}{\delta \mu}[\mu](x) = C_I - \frac{1}{2}\sum_{j \in J_I} \omega_j \left(a_j \cdot x\right)^2 \qquad (\text{for } x \in I) \tag{A.22}$$

for some $C_I \in \mathbb{R}$. Both sides are real-analytic on $\mathbb{S}^{d-1}$ and $I$ is open, so by the identity theorem this equality holds on $\mathbb{S}^{d-1}$.

By Lemma A.1(i), it follows that $\mu$ is absolutely continuous with a density $\rho$ which is a spherical polynomial of degree $\leqslant 2$; in particular, $\rho$ is real-analytic and not identically zero. Since the zero set of a nontrivial real-analytic function on $\mathbb{S}^{d-1}$ has null $\sigma_d$-measure, it follows that $\operatorname{supp}\mu = \mathbb{S}^{d-1}$.

Since $\operatorname{supp}\mu = \mathbb{S}^{d-1}$, the stationarity condition (2.8) holds on all of $\mathbb{S}^{d-1}$. Hence $g = \nabla H = 0$ on $\mathbb{S}^{d-1}$, and therefore $H$ is constant on $\mathbb{S}^{d-1}$. On the other hand, (A.22) already shows that $\frac{\delta E_\beta}{\delta \mu}[\mu]$ is the restriction to $\mathbb{S}^{d-1}$ of a quadratic polynomial. It follows that

$$\mathsf{v}_\vartheta = 2H - 2\frac{\delta E_\beta}{\delta \mu}[\mu]$$

is also the restriction to $\mathbb{S}^{d-1}$ of a quadratic polynomial. In particular, $\mathsf{v}_\vartheta$ is real-analytic on $\mathbb{S}^{d-1}$, contradicting the assumption. Therefore $\sigma_d(\operatorname{supp}\mu) = 0$.

Now assume that $\sigma$ is real-analytic and that $\mu$ is a strict SOPD Wasserstein critical point. If $\sigma_d(\operatorname{supp}\mu) > 0$, then the real-analytic vector field $g = \nabla H$ vanishes on a set of positive $\sigma_d$-measure. Hence $g \equiv 0$ on $\mathbb{S}^{d-1}$ by analyticity (Mityagin, 2020), and therefore $H$ is constant on $\mathbb{S}^{d-1}$. The same localization argument as in the proof of Theorem 3.2 then yields unit-norm tangent perturbations along which the corresponding Hessian values become arbitrarily small, contradicting (2.5).

PART II. ATOMICITY

Fix $\mu \in \mathcal{P}(\mathbb{S}^{d-1})$ and define

$$g_{\beta,\vartheta}(x) := \nabla\frac{\delta E_{\beta,\vartheta}}{\delta \mu}[\mu](x) = \int e^{\beta x \cdot y}\mathbf{P}_x^\perp y \, \mathrm{d}\mu(y) + \sum_{j \in [\![1,d]\!]} \omega_j \sigma(a_j \cdot x)\mathbf{P}_x^\perp a_j.$$

If $\mu$ is stationary, then $\operatorname{supp}\mu \subset \{x \in \mathbb{S}^{d-1}: g_{\beta,\vartheta}(x) = 0\}$, by (2.8).

Our goal is to show that, for a dense subset of parameters, all zeros of $g_{\beta,\vartheta}$ are nondegenerate (and thus isolated). To achieve this, we invoke the parametric transversality theorem (Lee, 2012, Thm. 6.35). This theorem guarantees the result provided that the map $(x, \beta, \vartheta) \mapsto g_{\beta,\vartheta}(x)$ is transverse to the zero section.

Explicitly, we need to verify that for all $(x, \beta, \vartheta)$ such that $g_{\beta,\vartheta}(x) = 0$, the differential map surjects onto the tangent space:

$$\operatorname{Im} D_{(\beta,\vartheta)}g_{\beta,\vartheta}(x) + \operatorname{Im} D_x g_{\beta,\vartheta}(x) = \mathsf{T}_x\mathbb{S}^{d-1}. \tag{A.23}$$

If (A.23) holds, the theorem implies that for almost every $(\beta, \vartheta)$, and hence for a dense set of parameters, the zeros of $x \mapsto g_{\beta,\vartheta}(x)$ are nondegenerate.

Assume first that $\sigma$ is real-analytic and $\sigma(s) \neq 0$ for all $s \neq 0$. For fixed $\mu$, the map $(x, \beta, \vartheta) \mapsto g_{\beta,\vartheta}(x)$ is then real-analytic on $\mathbb{S}^{d-1} \times \mathbb{R}_{>0} \times (\mathbb{R}^{d+1})^d$ where $\vartheta = (a_j, \omega_j)_{j \in [\![1,d]\!]}$ are the perceptron parameters. We restrict to the open dense parameter set where $a_j$ are linearly independent and $\omega_j \neq 0$ for all $j$. On this subset, for every $x \in \mathbb{S}^{d-1}$ there exists $j \in [\![1,d]\!]$ such that $a_j \cdot x \neq 0$, thus $\omega_j\sigma(a_j \cdot x) \neq 0$ thanks to the assumption on $\sigma$.

The differential with respect to $a_j$ evaluated at $h \in \mathbb{R}^d$ is given by

$$D_{a_j}g_{\beta,\vartheta}(x)[h] = \omega_j\left(\sigma'\left(a_j \cdot x\right)\left(x \cdot h\right)\mathbf{P}_x^\perp a_j + \sigma\left(a_j \cdot x\right)\mathbf{P}_x^\perp h\right).$$

By restricting to $h \in \mathsf{T}_x\mathbb{S}^{d-1}$ so that $\mathbf{P}_x^\perp h = h$ and $x \cdot h = 0$, we deduce that

$$D_{a_j}g_{\beta,\vartheta}(x)[h] = \omega_j\sigma\left(a_j \cdot x\right)h.$$

Since $\omega_j \sigma(a_j \cdot x) \neq 0$, the linear map $D_{a_j} g_{\beta,\vartheta}(x)$ is onto $\mathsf{T}_x \mathbb{S}^{d-1}$. In particular, (A.23) holds. By the parametric transversality theorem (Lee, 2012, Thm. 6.35), the set of parameters for which all zeros of $g_{\beta,\vartheta}$ are nondegenerate is dense. Moreover, since $\mathbb{S}^{d-1}$ is compact, this set of parameters is also open. Finally, since nondegenerate zeros are isolated by the inverse function theorem, the compactness of $\mathbb{S}^{d-1}$ guarantees that $g_{\beta,\vartheta}$ has only finitely many zeros.

Now assume $\sigma(s) = s_+$. Then $x \mapsto g_{\beta,\vartheta}(x)$ is smooth on each component $I$ of $\mathbb{S}^{d-1} \setminus \mathscr{Z}$. As before, assume $a_j$ linearly independent and $\omega_j \neq 0$. Fix a component $I$ with $J_I \neq \varnothing$ and let $x \in I$ with $g_{\beta,\vartheta}(x) = 0$. Choose any $j \in J_I$. Then for any $h \in \mathsf{T}_x \mathbb{S}^{d-1}$, the differential with respect to $a_j$ is given by

$$D_{a_j} g_{\beta,\vartheta}(x)[h] = \omega_j \sigma(a_j \cdot x) h = \omega_j (a_j \cdot x) h.$$

Since $a_j \cdot x > 0$ on $I$ and $\omega_j \neq 0$, this map is a scaled isometry, hence surjective onto $\mathsf{T}_x \mathbb{S}^{d-1}$. By the parametric transversality theorem, for a dense set of parameters, the zeros in $I$ are nondegenerate and thus form a discrete set.

To handle the boundaries, we consider the submanifolds $F$ defined by the intersection of some hyperplanes $\{x : a_k \cdot x = 0\}$ inside $\mathcal{A} := \bigcup_j \{x : a_j \cdot x > 0\}$. Since the restriction of $g_{\beta,\vartheta}$ to $F$ takes values in the larger space $\mathsf{T}\mathbb{S}^{d-1}$, we consider the projected field $g^F(x) := \mathbf{P}^{\perp}_{\mathsf{T}_x F} g_{\beta,\vartheta}(x)$ for $x \in F$. Note that if $g_{\beta,\vartheta}(x) = 0$, then necessarily $g^F(x) = 0$. We apply transversality to the map $g^F : F \to \mathsf{T}F$. Let $x \in F$ be a zero of $g^F$. Since $x \in \mathcal{A}$, there exists $j$ such that $a_j \cdot x > 0$. For any $h \in \mathsf{T}_x F$, the differential with respect to $a_j$ acts as

$$D_{a_j} g^F(x)[h] = \mathbf{P}^{\perp}_{\mathsf{T}_x F} (\omega_j (a_j \cdot x) h) = \omega_j (a_j \cdot x) h.$$

This map surjects onto $\mathsf{T}_x F$. Thus, for a dense set of parameters, the zeros of $g^F$ are isolated in $F$. By inclusion, the zeros of $g_{\beta,\vartheta}$ in $F$ are also isolated.

Since $\mathscr{Z}$ consists of finitely many hyperplanes, $\mathcal{A}$ is the finite union of such $I$ and $F$. Consequently, the set of zeros of $g_{\beta,\vartheta}$ in $\mathcal{A}$ is a finite union of discrete sets. If $\mu$ is stationary, then $\operatorname{supp}\mu \cap \mathcal{A}$ is countable. $\qquad\square$

*Remark* A.5. Theorem 3.3(**ii**) does not strictly preclude the existence of non-atomic stationary measures; rather, it implies that such a measure $\mu$ can only be stationary if the parameters lie in the exceptional (non-generic) set associated with $\mu$. A concrete example is the activation $\sigma(s) = s$, which yields a quadratic potential $\mathsf{v}_\vartheta(x) = \sum_j \omega_j (a_j \cdot x)^2$. If a stationary measure $\mu$ has support with non-empty interior, the stationarity condition forces the attention field to be locally (and thus globally) quadratic. By Lemma A.1(**i**), this implies $\mu$ admits a smooth density (a spherical polynomial of degree $\leqslant 2$). Since such a measure is not finitely supported, the parameters allowing it must necessarily lie in the non-generic set specific to $\mu$.

## A.4. Proofs of Theorem 3.5 and Corollary 3.6

*Proof of Theorem 3.5.* We prove a more general estimate for an arbitrary parameter $\lambda \in (0,1)$, from which the theorem follows by setting $\lambda = 1/2$.

Restricting the energy functional $\mathsf{E}_{\beta,\vartheta}$ to atomic measures of the form $\nu = \sum_{i \in [\![1,N]\!]} m_i \delta_{\phi_i}$ (where $m_i$ are fixed as in (3.1)), it becomes a function of the angular coordinates $\phi = (\phi_1, \ldots, \phi_N) \in (\mathbb{R}/2\pi\mathbb{Z})^N$:

$$\mathsf{E}_{\beta,\vartheta}(\phi) = \frac{1}{2\beta} \sum_{i,j \in [\![1,N]\!]} m_i m_j \mathsf{K}_\beta(\phi_i - \phi_j) + \frac{1}{2} \sum_{i \in [\![1,N]\!]} m_i (\mathsf{v}_\vartheta \circ x)(\phi_i).$$

Since $\mu$ is a SOPD Wasserstein critical point, it satisfies (2.8) and (2.4). The Wasserstein Hessian at $\mu$ coincides with the Euclidean Hessian of $\mathsf{E}_{\beta,\vartheta}$ in the variables $\phi$, evaluated at the configuration $\theta$. In particular, for every $\xi \in \mathbb{R}^N$,

$$\nabla^2 \mathsf{E}_{\beta,\vartheta}(\theta)[\xi, \xi] = \operatorname{Hess}_\mu \mathsf{E}_{\beta,\vartheta}(\xi, \xi) \geqslant 0. \tag{A.24}$$

Thus, the matrix $\nabla^2 \mathsf{E}_{\beta,\vartheta}(\theta)$, with entries defined by

$$\frac{\partial^2 \mathsf{E}_{\beta,\vartheta}}{\partial \phi_i \partial \phi_j}(\theta) = \begin{cases} -\frac{1}{\beta} m_i m_j \mathsf{K}_\beta''(\theta_i - \theta_j), & \text{for } i \neq j, \\ \frac{1}{\beta} m_i \sum_{k \neq i} m_k \mathsf{K}_\beta''(\theta_i - \theta_k) + \frac{1}{2} m_i \partial_\theta^2 (\mathsf{v}_\vartheta \circ x)(\theta_i), & \text{for } i = j, \end{cases}$$

is positive semi-definite.

Fix $\lambda \in (0,1)$, and consider a cluster of $n \in [\![2, N]\!]$ atoms satisfying the distance condition

$$\min_{k \in \mathbb{Z}} |\theta_i - \theta_j + 2\pi k| \leqslant \frac{\lambda}{\sqrt{\beta}} \qquad \text{for all } i, j \in [\![1, n]\!]. \tag{A.25}$$

We evaluate the quadratic form $\nabla^2 \mathsf{E}_{\beta, \vartheta}(\boldsymbol{\theta})[\xi, \xi]$ along the vector $\xi \in \mathbb{R}^N$ defined by

$$\xi_i = \frac{1}{m_i} \quad (i < n), \qquad \xi_n = -\frac{n-1}{m_n}, \qquad \xi_i = 0 \quad (i > n).$$

Note that $\sum_{i \in [\![1,n]\!]} m_i \xi_i = 0$. We split

$$\nabla^2 \mathsf{E}_{\beta, \vartheta}(\boldsymbol{\theta})[\xi, \xi] = \mathcal{T}_{\mathrm{in}} + \mathcal{T}_{\mathrm{out}} + \mathcal{T}_{\vartheta}, \tag{A.26}$$

where

$$\mathcal{T}_{\mathrm{in}} := \frac{1}{\beta} \sum_{1 \leqslant i < j \leqslant n} m_i m_j \, \mathsf{K}_{\beta}'' (\theta_i - \theta_j)(\xi_i - \xi_j)^2, \tag{A.27}$$

$$\mathcal{T}_{\mathrm{out}} := \frac{1}{\beta} \sum_{i \in [\![1,n]\!]} \sum_{k > n} m_i m_k \, \mathsf{K}_{\beta}'' (\theta_i - \theta_k) \xi_i^2, \tag{A.28}$$

$$\mathcal{T}_{\vartheta} := \frac{1}{2} \sum_{i \in [\![1,n]\!]} m_i \, \partial_{\theta}^2 (\mathsf{v}_{\vartheta} \circ x)(\theta_i) \xi_i^2. \tag{A.29}$$

Using the above choice of $\xi_i$ and the identity

$$\sum_{1 \leqslant i < j \leqslant n} m_i m_j (\xi_i - \xi_j)^2 = \sum_{i \in [\![1,n]\!]} m_i \sum_{j \in [\![1,n]\!]} m_j \xi_j^2 - \left( \sum_{i \in [\![1,n]\!]} m_i \xi_i \right)^2,$$

we can estimate

$$\mathcal{T}_{\mathrm{in}} \leqslant \frac{1}{\beta} \max_{1 \leqslant i < j \leqslant n} \mathsf{K}_{\beta}'' (\theta_i - \theta_j) \sum_{i \in [\![1,n]\!]} m_i \sum_{j \in [\![1,n]\!]} m_j \xi_j^2.$$

Using (A.25) and the fact that $\lambda/\sqrt{\beta} \sim \lambda \theta_c(\beta)$ for large $\beta$, we can directly apply (A.6) from Lemma A.3. For all $\beta$ sufficiently large, this yields

$$\max_{1 \leqslant i < j \leqslant n} \mathsf{K}_{\beta}'' (\theta_i - \theta_j) \leqslant \mathsf{K}_{\beta}'' \left( \frac{\lambda}{\sqrt{\beta}} \right) \leqslant -\frac{1}{2} \left( 1 - \lambda^2 \right) \beta \, e^{\beta - \lambda^2/2}.$$

Substituting this bound into the definition of $\mathcal{T}_{\mathrm{in}}$, we obtain

$$\mathcal{T}_{\mathrm{in}} \leqslant -\frac{e^{\beta - \frac{\lambda^2}{2}} \left( 1 - \lambda^2 \right)}{2} \sum_{i \in [\![1,n]\!]} m_i \sum_{j \in [\![1,n]\!]} m_j \xi_j^2. \tag{A.30}$$

Second, from (A.7) we obtain, for large enough $\beta$,

$$\mathcal{T}_{\mathrm{out}} \leqslant 2 e^{\beta - \frac{3}{2}} \sum_{k > n} m_k \sum_{i \in [\![1,n]\!]} m_i \xi_i^2. \tag{A.31}$$

Finally, since $\sigma$ is globally Lipschitz and $\|x\| = \|x'\| = \|x''\| = 1$, we can compute for a.e. $\theta$:

$$\frac{1}{2} \partial_{\theta}^2 (\mathsf{v}_{\vartheta} \circ x)(\theta) = \sum_{j \in [\![1,2]\!]} \omega_j \left( \sigma' (a_j \cdot x(\theta)) (a_j \cdot x'(\theta))^2 + \sigma (a_j \cdot x(\theta)) a_j \cdot x''(\theta) \right)$$

$$\leqslant \sum_{j \in [\![1,2]\!]} |\omega_j| \left( \|\sigma\|_{C^{0,1}} \|a_j\|^2 + (|\sigma(0)| + \|\sigma\|_{C^{0,1}} \|a_j\|) \|a_j\| \right) =: C_{\vartheta}.$$

Therefore

$$\mathcal{T}_\vartheta \leqslant C_\vartheta \sum_{i \in [\![1,n]\!]} m_i \xi_i^2. \tag{A.32}$$

Combining (A.26), (A.30), (A.31), and (A.32), we obtain an upper bound for the Hessian:

$$\nabla^2 \mathsf{E}_{\beta,\vartheta}\left(\boldsymbol{\theta}\right)[\xi,\xi] \leqslant \left[-\frac{e^{\beta - \frac{\lambda^2}{2}}\left(1 - \lambda^2\right)}{2} \sum_{i \in [\![1,n]\!]} m_i + 2e^{\beta - \frac{3}{2}} \sum_{k>n} m_k + C_\vartheta\right] \sum_{j \in [\![1,n]\!]} m_j \xi_j^2. \tag{A.33}$$

For the second order condition (A.24) to hold, the right-hand side cannot be negative. Therefore:

$$\frac{e^{\beta - \frac{\lambda^2}{2}}\left(1 - \lambda^2\right)}{2} \sum_{i \in [\![1,n]\!]} m_i \leqslant 2e^{\beta - \frac{3}{2}}\left(1 - \sum_{i \in [\![1,n]\!]} m_i\right) + C_\vartheta.$$

Solving for $\sum_{i \in [\![1,n]\!]} m_i$, we get

$$\sum_{i \in [\![1,n]\!]} m_i \leqslant \frac{2e^{\beta - \frac{3}{2}} + C_\vartheta}{2e^{\beta - \frac{3}{2}} + \frac{e^{\beta - \frac{\lambda^2}{2}}(1-\lambda^2)}{2}} = \frac{4}{4 + e^{\frac{3}{2} - \frac{\lambda^2}{2}}\left(1 - \lambda^2\right)} + O\left(e^{-\beta}\right).$$

Setting $\lambda = 1/2$, this bound yields (3.3).

For the second part, assume that $\mu$ is itself a single cluster satisfying the pairwise distance condition (so $n = N$ and $\sum_{k>n} m_k = 0$). Then (A.33) gives:

$$0 \leqslant \nabla^2 \mathsf{E}_{\beta,\vartheta}\left(\boldsymbol{\theta}\right)[\xi,\xi] \leqslant \left[-\frac{e^{\beta - \frac{\lambda^2}{2}}\left(1 - \lambda^2\right)}{2} + C_\vartheta\right] \sum_{j \in [\![1,n]\!]} m_j \xi_j^2,$$

hence necessarily

$$C_\vartheta \geqslant \frac{e^{\beta - \frac{\lambda^2}{2}}\left(1 - \lambda^2\right)}{2}.$$

With $\lambda = 1/2$ this reads

$$C_\vartheta \geqslant \frac{3}{8} e^{\beta - \frac{1}{8}}.$$

Under the standing assumptions $\|\sigma\|_{C^{0,1}} = 1$ and $\sigma(0) = 0$, we have $C_\vartheta = 2 \sum_{j \in [\![1,2]\!]} |\omega_j| \cdot \|a_j\|^2$. Therefore, if

$$|\omega_1| \cdot \|a_1\|^2 + |\omega_2| \cdot \|a_2\|^2 < \frac{3}{16} e^{-\frac{1}{8}} < 0.16547,$$

then $C_\vartheta < \frac{3}{8} e^{-\frac{1}{8}} \leqslant \frac{3}{8} e^{\beta - \frac{1}{8}}$ for all $\beta > 0$. This excludes $\operatorname{supp} \mu$ being a single such cluster.

$\square$

*Remark* A.6. The restriction of Theorem 3.5 to $d = 2$ is technical: our proof uses the one-dimensional angular parametrization of $\mathbb{S}^1$ and the scalar concavity estimate in Lemma A.3; extending it to $d \geqslant 3$ would require a higher-dimensional Hessian estimate on small geodesic caps.

*Proof of Corollary 3.6.* Let $\Lambda_\beta$ denote the exact mass bound derived in Theorem 3.5 for a cluster of diameter $\frac{1}{2\sqrt{\beta}}$, i.e.,

$$\Lambda_\beta := 0.5742 + O(e^{-\beta}).$$

We define a covering of the support of $\mu$. For each of the arcs $I_j$ of length $|I_j| \leqslant L$, we partition it into $K_j$ disjoint sub-intervals $J_{j,1}, \ldots, J_{j,K_j}$, each of length at most $1/(2\sqrt{\beta})$. The minimal number of such intervals required is

$$K_j := \max\left\{1, \left\lceil 2|I_j|\sqrt{\beta}\right\rceil\right\} \leqslant 1 + 2|I_j|\sqrt{\beta} \leqslant 1 + 2L\sqrt{\beta}.$$

Consider any such sub-interval $J = J_{j,k}$. Let $n_\varepsilon(J) := \#\{i : \theta_i \in J, \ m_i \geqslant \varepsilon\}$. We distinguish cases based on the number of atoms in $J$:

- If $J$ contains at least two atoms, they form a cluster satisfying the pairwise distance condition of Theorem 3.5 (since the diameter of $J$ is $\leqslant 1/(2\sqrt{\beta})$). Thus, $\mu(J) \leqslant \Lambda_\beta$.

- If $J$ contains 0 or 1 atom, then obviously $n_\varepsilon(J) \leqslant 1$.

Combining these, if $\varepsilon \leqslant \Lambda_\beta$, we have $1 \leqslant \Lambda_\beta/\varepsilon$. Therefore, in the first case $n_\varepsilon(J) \cdot \varepsilon \leqslant \mu(J) \leqslant \Lambda_\beta$, and in the second case $n_\varepsilon(J) \leqslant 1 \leqslant \Lambda_\beta/\varepsilon$. Hence, for any sub-interval, $n_\varepsilon(J) \leqslant \Lambda_\beta/\varepsilon$.

Since every atom with mass $\geqslant \varepsilon$ belongs to at least one sub-interval $J_{j,k}$, we have

$$N_\varepsilon \leqslant \sum_{j \in [\![1,M]\!]} \sum_{k \in [\![1,K_j]\!]} n_\varepsilon(J_{j,k}) \leqslant \sum_{j \in [\![1,M]\!]} K_j \frac{\Lambda_\beta}{\varepsilon}.$$

Substituting the bound for $K_j$, we obtain:

$$N_\varepsilon \leqslant M \left(1 + 2L\sqrt{\beta}\right) \frac{\Lambda_\beta}{\varepsilon}.$$

Finally, if $\varepsilon > \Lambda_\beta$, we use the trivial bound $N_\varepsilon \leqslant 1/\varepsilon$. For $\beta$ large enough such that $M \left(1 + 2L\sqrt{\beta}\right) \Lambda_\beta \geqslant 1$, the bound in the statement is larger than $1/\varepsilon$ and thus remains valid. $\square$

### A.5. Proof of Proposition 3.7

*Proof of Proposition 3.7.* Since $e^{\beta x \cdot y} \leqslant e^\beta$ on $\mathbb{S}^{d-1}$, we have for any $\mu$,

$$\mathsf{E}_{\beta,\vartheta}[\mu] = \frac{1}{2\beta} \iint e^{\beta x \cdot y} \, \mathrm{d}\mu(x) \, \mathrm{d}\mu(y) + \frac{1}{2} \int \mathsf{v}_\vartheta \, \mathrm{d}\mu \leqslant \frac{e^\beta}{2\beta} + \frac{1}{2} \max_{x \in \mathbb{S}^{d-1}} \mathsf{v}_\vartheta(x),$$

and equality holds iff $\mu = \delta_x$ with $x \in \arg\max_{y \in \mathbb{S}^{d-1}} \sum_{j \in [\![1,d]\!]} \omega_j \, \varphi(a_j \cdot y)$. This proves the characterization of global maximizers.

Existence of the minimizer follows from compactness of $\mathcal{P}(\mathbb{S}^{d-1})$ and continuity of the integrands. To see uniqueness, note that the kernel $e^{\beta x \cdot y}$ is conditionally strictly positive definite: for any finite signed measure $\nu$ with $\nu(\mathbb{S}^{d-1}) = 0$,

$$\iint e^{\beta x \cdot y} \, \mathrm{d}\nu(x) \, \mathrm{d}\nu(y) = \sum_{k \geqslant 1} \lambda_k(\beta) \sum_{\ell \in [\![1,N_k]\!]} \left( \int Y_{k\ell} \, \mathrm{d}\nu \right)^2 > 0 \quad \text{if } \nu \neq 0.$$

Thus $\mathsf{E}_\beta$ is strictly convex on $\mathcal{P}(\mathbb{S}^{d-1})$, see (Bilyk et al., 2022, Prop. 2.1 & Thm. 4.1). Adding the linear term $\frac{1}{2} \int \mathsf{v}_\vartheta \, \mathrm{d}\mu$ preserves strict convexity; thus the minimizer is unique.

Finally, $\mathsf{E}_\beta[R_\# \mu] = \mathsf{E}_\beta[\mu]$ for all $R \in O(d)$, and if moreover $Ra_j = a_j$ for all $j$, then

$$\mathsf{v}_\vartheta(Rx) = \sum_{j \in [\![1,d]\!]} \omega_j \, \varphi(a_j \cdot Rx) = \sum_{j \in [\![1,d]\!]} \omega_j \, \varphi(Ra_j \cdot x) = \mathsf{v}_\vartheta(x).$$

Hence $\mathsf{E}_{\beta,\vartheta}[R_\# \mu] = \mathsf{E}_{\beta,\vartheta}[\mu]$ for all such $R$, and by uniqueness $R_\# \mu_* = \mu_*$. $\square$

*Remark A.7.* It is clear that the minimizer $\mu_*$ is SOPD. Under additional curvature from the perceptron potential, one can ensure strictness. For $d = 2$, (A.18) yields for all $\xi \in \mathsf{T}_\mu \mathcal{P}(\mathbb{S}^1)$:

$$
\begin{aligned}
\mathrm{Hess}_\mu \mathsf{E}_{\beta,\vartheta}(\xi,\xi) &= \int \left[ \frac{1}{\beta} \left( \mathsf{K}''_\beta * \mu \right)(\theta) + \frac{1}{2} \partial_\theta^2 \left( \mathsf{v}_\vartheta \circ x \right)(\theta) \right] \xi(\theta)^2 \, \mathrm{d}\mu(\theta) - \frac{1}{\beta} \iint \mathsf{K}''_\beta(\theta - \phi) \, \xi(\theta) \, \xi(\phi) \, \mathrm{d}\mu(\theta) \, \mathrm{d}\mu(\phi) \\
&\geqslant \left( \frac{1}{2} \inf_{\theta \in \mathrm{supp}\, \mu} \partial_\theta^2 \left( \mathsf{v}_\vartheta \circ x \right)(\theta) - \frac{\|\mathsf{K}''_\beta\|_\infty}{\beta} \right) \|\xi\|_{L^2(\mu)}^2,
\end{aligned}
$$

where we used $\iint \left( \xi(\theta) - \xi(\phi) \right)^2 \, \mathrm{d}\mu(\theta) \, \mathrm{d}\mu(\phi) = 2\|\xi\|_{L^2(\mu)}^2 - 2 \left( \int \xi \, \mathrm{d}\mu \right)^2 \leqslant 2\|\xi\|_{L^2(\mu)}^2$.

Consequently, a sufficient condition for $\mu$ to be strict SOPD is: there exists $\kappa > 0$ such that

$$\partial_\theta^2 \left(\mathsf{v}_\vartheta \circ x\right)(\theta) \geqslant \frac{2\|\mathsf{K}_\beta''\|_\infty}{\beta} + 2\kappa \qquad (\theta \in \operatorname{supp}\mu). \tag{A.34}$$

For $\mathsf{K}_\beta(\phi) = e^{\beta\cos\phi}$ one has $\mathsf{K}_\beta''(\phi) = \left(\beta^2\sin^2\phi - \beta\cos\phi\right)e^{\beta\cos\phi}$, hence $\|\mathsf{K}_\beta''\|_\infty = \beta e^\beta$. Thus, for $\mathsf{v}_\vartheta$ as in (2.7), condition (A.34) simplifies to

$$\sum_{j\in[\![1,2]\!]} \omega_j \left(\sigma'\left(a_j \cdot x(\theta)\right)\left(a_j \cdot x'(\theta)\right)^2 + \sigma\left(a_j \cdot x(\theta)\right)a_j \cdot x''(\theta)\right) \geqslant e^\beta + \kappa, \qquad (\theta \in \operatorname{supp}\mu).$$

An analogous sufficient condition holds for $d \geqslant 3$, replacing $\partial_\theta^2\left(\mathsf{v}_\vartheta \circ x\right)$ with the minimum eigenvalue of $\nabla^2\mathsf{v}_\vartheta$, and $\|\mathsf{K}_\beta''\|_\infty$ with the constant $C_{\beta,d} < \infty$ such that $\operatorname{Hess}_\mu\mathsf{E}_\beta(\xi,\xi) \geqslant -C_{\beta,d}\|\xi\|_{L^2(\mu)}^2$ for all $\mu$ and $\xi \in \mathsf{T}_\mu\mathcal{P}(\mathbb{S}^{d-1})$.

## B. Normalized self-attention

As noted in Section 2, the normalized attention field can be viewed as a weighted gradient. The sparsity results obtained for the unnormalized case extend to this setting, provided we impose a mild non-degeneracy condition on the perceptron weights.

**Proposition B.1.** *Let $d \geqslant 2$, $\beta > 0$, fix $\sigma(s) = s_+$, and let $\mu \in \mathcal{P}(\mathbb{S}^{d-1})$ be a stationary measure for (2.9), i.e., satisfying (2.10). Assume that $(\omega_j, a_j)_j$ satisfy the following non-degeneracy condition: for every index subset $J \subseteq [\![1,d]\!]$, the matrix*

$$M := \sum_{j\in J} \omega_j\, a_j a_j^\top - \sum_{j\notin J} \omega_j\, a_j a_j^\top \tag{B.1}$$

*is not a scalar multiple of the identity. Then $\sigma_d(\operatorname{supp}\mu) = 0$; in particular, $\mu$ is singular with respect to $\sigma_d$. Moreover, if $d = 2$, then $\mu$ is purely atomic with finitely many atoms.*

*Proof of Proposition B.1.* Because $\frac{\delta\mathsf{E}_\beta}{\delta\mu}[\mu]$ is strictly positive and real-analytic in $\mathbb{S}^{d-1}$, its logarithm is well-defined and real-analytic. On the other hand, on each connected component $I$ of the set $\mathbb{S}^{d-1} \setminus \mathscr{Z}$ we have

$$\mathsf{v}_\vartheta(x) = \sum_{j\in[\![1,d]\!]} \omega_j\left(a_j \cdot x\right)_+^2 = \sum_{j\in J_I} \omega_j\left(a_j \cdot x\right)^2 \quad \text{for } x \in I,$$

where $\mathscr{Z}$ and $J_I$ are defined in (3.5) and (3.6). Therefore

$$H_{\log} := \log\left(\frac{\delta\mathsf{E}_\beta}{\delta\mu}[\mu]\right) + \frac{1}{2}\mathsf{v}_\vartheta, \qquad g_{\log} := \nabla H_{\log}$$

are real-analytic on $I$. By (2.10), we get $\operatorname{supp}\mu \subset \{g_{\log} = 0\}$.

(a) Suppose, for contradiction, that $\sigma_d(\operatorname{supp}\mu \cap I) > 0$ for some component $I$. As $g_{\log}$ is real-analytic on $I$ and vanishes on a set of positive measure, we have $g_{\log} \equiv 0$ on $I$, hence $H_{\log}$ is constant on $I$: there exists $C_I \in \mathbb{R}$ such that

$$\log\left(\frac{\delta\mathsf{E}_\beta}{\delta\mu}[\mu](x)\right) = C_I - \frac{1}{2}\sum_{j\in J_I}\omega_j\left(a_j \cdot x\right)^2, \qquad x \in I.$$

Define the global real-analytic vector field

$$\widetilde{g}_{\log}(x) := \nabla\left(\log\left(\frac{\delta\mathsf{E}_\beta}{\delta\mu}[\mu](x)\right) + \frac{1}{2}\sum_{j\in J_I}\omega_j\left(a_j \cdot x\right)^2\right).$$

Since $\widetilde{g}_{\log} = g_{\log} \equiv 0$ on $I$, the identity theorem for real-analytic functions on $\mathbb{S}^{d-1}$ implies $\widetilde{g}_{\log} \equiv 0$ on $\mathbb{S}^{d-1}$. Therefore

$$\frac{\delta\mathsf{E}_\beta}{\delta\mu}[\mu](x) = \exp\left(C_I - \frac{1}{2}\sum_{j\in J_I}\omega_j\left(a_j \cdot x\right)^2\right), \qquad x \in \mathbb{S}^{d-1}. \tag{B.2}$$

The right-hand side of (B.2) is even, so $\mu(A) = \mu(-A)$ for every Borel set $A \subset \mathbb{S}^{d-1}$, by Lemma A.1**(ii)**. Thus $\operatorname{supp} \mu$ meets the antipodal component $I'$ of $I$, whose active set is $J_{I'} = \llbracket 1, d \rrbracket \setminus J_I$. Repeating the argument on $I'$ yields

$$\frac{\delta \mathsf{E}_\beta}{\delta \mu}[\mu](x) = \exp\left(C_{I'} - \frac{1}{2} \sum_{j \in J_{I'}} \omega_j \left(a_j \cdot x\right)^2\right), \qquad x \in \mathbb{S}^{d-1}.$$

Equating the two global expressions gives, for all $x \in \mathbb{S}^{d-1}$,

$$\sum_{j \in J_I} \omega_j \left(a_j \cdot x\right)^2 - \sum_{j \notin J_I} \omega_j \left(a_j \cdot x\right)^2 \equiv 2\left(C_I - C_{I'}\right). \tag{B.3}$$

This means that $x^\top M x$ is constant on $\mathbb{S}^{d-1}$ for $M$ defined as in (B.1), which forces $M$ to be a scalar multiple of the identity, contradicting our assumption. Thus $\sigma_d(\operatorname{supp} \mu \cap I) = 0$ for every $I$. Since $\sigma_d(\partial I) = 0$, we conclude $\sigma_d(\operatorname{supp} \mu) = 0$.

(b) Let $d = 2$ and suppose $\operatorname{supp} \mu$ is infinite. Then $\operatorname{supp} \mu \cap I$ has an accumulation point in $\mathbb{S}^1$ for some $I$. The same identity theorem applied to $\widetilde{g}_{\log}$ yields (B.2) and (B.3), leading to the same contradiction. By compactness of $\mathbb{S}^1$, the support must be finite. $\qquad \square$

*Remark* B.2. The argument above also applies to normalized attention with a symmetric nonsingular matrix $B \in \mathbb{R}^{d \times d}$ by replacing $\mathsf{E}_\beta$ with $\mathsf{E}_B[\mu] := \frac{1}{2} \iint e^{x^\top B y} \, \mathrm{d}\mu(x) \, \mathrm{d}\mu(y)$. Since $\frac{\delta \mathsf{E}_B}{\delta \mu}[\mu]$ is strictly positive and real-analytic, the inclusion $\operatorname{supp} \mu \subset \{\nabla H_{\log} = 0\}$ still leads to the global identity (B.2). By Remark A.2, this implies that $\mu(A) = \mu(-A)$ for every Borel set $A \subset \mathbb{S}^{d-1}$, which yields the same contradiction via (B.3). A similar argument extends Theorems 3.1 and 3.3 to $\mathsf{E}_B$ by removing the exponential from the analogous identities.

## C. Supplementary simulations

We now present numerical experiments for higher dimensions, specifically $d \in \{4, 5, 7, 10, 20, 50\}$. We update the cluster identification rule so that particles are grouped if their pairwise geodesic distance is at most $\min\{1/(2\sqrt{\beta}), \pi/(2d)\}$. This heuristic choice accounts for concentration of measure on high-dimensional spheres, where random points tend to be nearly orthogonal.

Figure 11 and Figure 12 summarize the cluster counts and their respective masses across dimensions $d \in \{4, 5, 7, 10, 20, 50\}$. Notably, we observe empirically that the cluster masses in these higher dimensions remain strictly below the same numerical upper bound derived for $d = 2$ in Theorem 3.5.

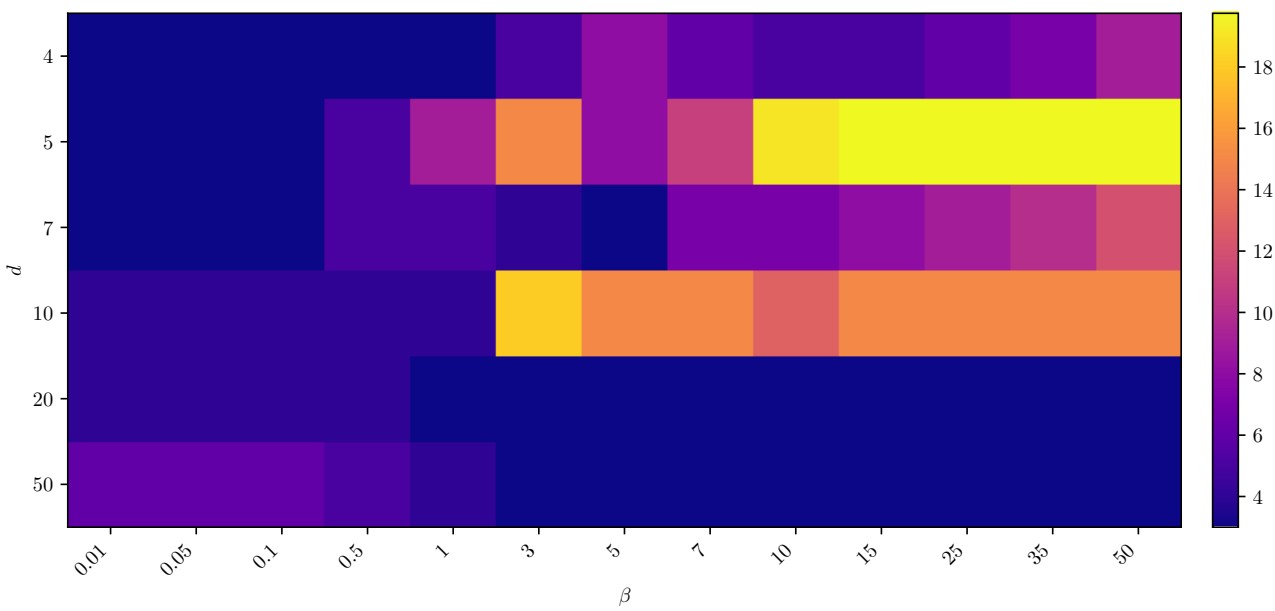

*Figure 11.* Number of clusters at convergence across dimensions $d \in \{4, 5, 7, 10, 20, 50\}$ for gradient descent using unnormalized self-attention with a ReLU perceptron.

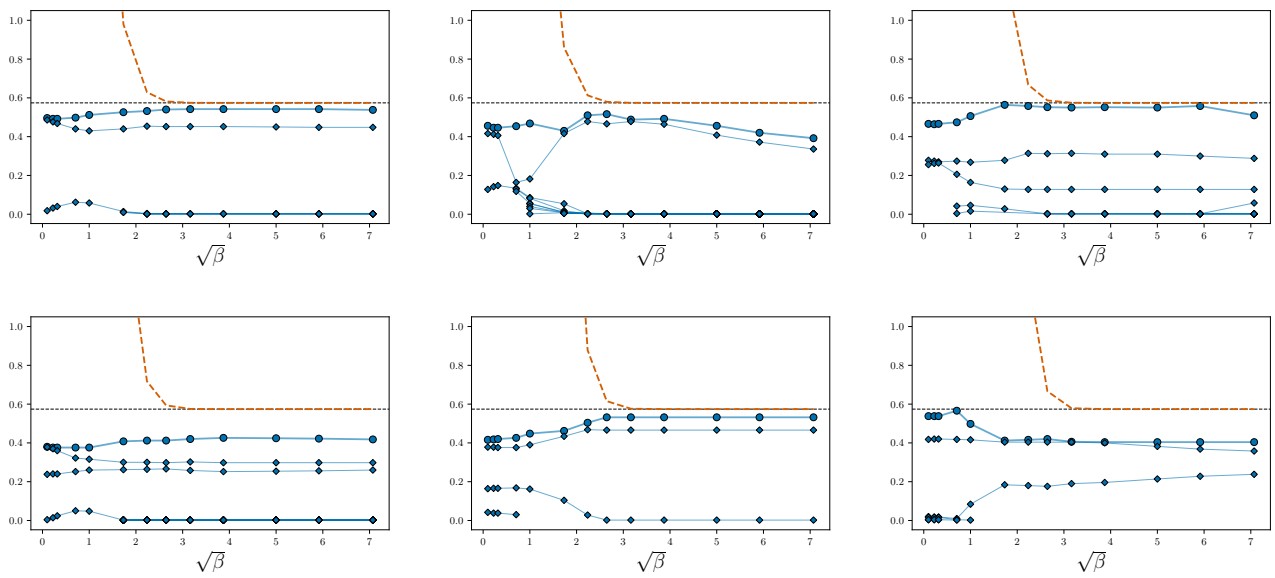

*Figure 12.* Cluster masses (in blue, the largest being the thickest) at final time across $\sqrt{\beta}$. The horizontal and red dashed lines represent the numerical term and the full upper bound in (3.3), respectively. **Top row:** Gradient descent with ReLU ($d = 4, 5, 7$). **Bottom row:** Same setup for higher dimensions ($d = 10, 20, 50$). (See Section 4 for setup).

