# OpenReview forum: "Perceptrons and Localization of Attention’s Mean-Field Landscape"
_ICML.cc/2026/Conference — ICML 2026 spotlight_

### Official Review · Reviewer_ydQ2 · 2026-02-22

**Soundness:** 3
**Presentation:** 3
**Significance:** 4
**Originality:** 3
**Overall Recommendation:** 5
**Confidence:** 2

**Summary:**

This paper studies the mean-field dynamics of Transformers on the unit sphere and proves that stationary measures are generically atomic. The results show that, in widely and deeply scaled regimes, perceptron blocks induce an intrinsic clustering tendency in the representation space.

**Compliance With Llm Reviewing Policy:**

Affirmed.

**Final Justification:**

I would keep a score of 5, as this paper addresses an important and timely question: obtaining theoretical insights into the behavior of Transformers.

**Key Questions For Authors:**

Please answer questions in (1) ~ (3).

**Limitations:**

The paper does not include an explicit discussion of its limitations. While Section 2.4 mentions modeling simplifications, it would be important to more clearly discuss the limitations of the analysis.

**Strengths And Weaknesses:**

The paper is technically sound, well organized, and easy to read. It addresses an important and timely question: obtaining theoretical insights into the behavior of Transformers. In particular, the paper shows that in the widely and deeply scaled regime, Transformers equipped with perceptron blocks exhibit an intrinsic clustering tendency, in the sense that stationary measures are generically atomic. This provides a fundamentally new understanding of the representation geometry of Transformers and constitutes a novel and original contribution.
The following are questions and suggestions for clarification :


(1) Dependence on the sphere $S^{d−1}$ :
The analysis is conducted entirely on the unit sphere $S^{d-1}$, motivated by layer normalization. However, it is not entirely clear to what extent this assumption is essential.
For example, there are existing works where attention mechanisms are studied without explicit projection onto the sphere, or on more general domains. In such cases, do similar conclusions hold? In particular, is the atomicity of stationary measures fundamentally tied to the spherical geometry, or is it expected to hold more generally on other compact domains?
A discussion on whether the results extend beyond $S^{d-1}$, or which parts of the proof rely critically on the spherical geometry, would significantly strengthen the paper. If the assumption of $S^{d-1}$ is mainly technical rather than essential, it would be helpful to explicitly discuss this point as a limitation of the current analysis, since the paper does not currently include a clear discussion of its limitations.

(2) Implications for continuous regression
This is not necessarily a limitation of the paper itself, but rather an implication of the results that deserves further discussion.
The paper suggests that Transformer dynamics with perceptrons intrinsically favor atomic stationary measures, i.e., clustered representations.
This raises an interesting question: does this clustering tendency suggest an intrinsic limitation of Transformers for learning continuous mappings, such as those arising in scientific machine learning or operator learning for PDEs, where the target mappings are inherently continuous? Even if this question is beyond the scope of the current paper, some discussion on how the present results relate to the use of Transformers in continuous regression settings would be valuable.

(3) Question on the derivation of equation (2.6)
In standard Transformers, the perceptron (MLP) and attention blocks are composed sequentially.
However, in equation (2.6), the mean-field PDE formulation results in a velocity field where the attention and perceptron contributions appear as a sum. Could the authors clarify why the composition of attention and MLP blocks leads to an additive structure at the level of the mean-field dynamics? Some intuition or derivation connecting the sequential architecture to this additive velocity field would improve clarity.

---

> ### Author Rebuttal · Authors · 2026-03-30
>
> **First Question.** We agree that it is important to clarify which parts of our analysis are fundamentally tied to the sphere and which could be generalized.
>
> The local analyticity and unique-continuation mechanisms are independent of the spherical geometry. Given any real-analytic manifold $\\mathcal M$, as long as the ReLU activation pattern induces a finite partition of $\\mathcal M$ into regions where the active set is fixed, the stationary drift remains real-analytic, yielding a local analytic identity if the support accumulates.
>
> However, our results regarding finite atomicity on $\\mathbb{S}^{1}$ and singularity on $\\mathbb{S}^{d-1}$ (for $d>2$) depend heavily on the sphere. Specifically, recovering global information on the measure $\\mu$ from the local identity requires the spectral structure of the isotropic kernel $e^{\\beta x\\cdot y}$ on $\\mathbb{S}^{d-1}$ (via spherical harmonics, cf. Lemma A.1. This inversion step, which allows us to propagate the contradiction, does not currently have a known analogue for more general domains.
>
> We will revise the paper to state this more clearly.
>
> **Second Question.** Our analysis focuses on stationary points as they govern the infinite-time behavior of the dynamics. While we prove that these states are atomic even in the presence of perceptrons, previous work has shown that the typical evolution of Transformers is extremely slow [1,2]. (Also note that by Cauchy-Lipschitz, an initial density remains a density when propagated by the Transformer flow over all finite time intervals.) This suggests that the empirical success of Transformers in continuous regression is likely due to a lack of global convexity/concavity, which prevents the representations from losing continuity over practical timescales, even if they are destined to cluster in the limit. Our results show that the inclusion of perceptrons does not fundamentally alter this long-term clustering tendency observed in simpler settings.
>
> [1] Geshkovski, B., Koubbi, H., Polyanskiy, Y., and Rigollet, P. Dynamic metastability in the self-attention model, 2024.
>
> [2] Bruno, G., Pasqualotto, F., and Agazzi, A. Emergence of meta-stable clustering in mean-field transformer models. In International Conference on Learning Representations, 2025.
>
> **Third Question.** The additive structure in equation (2.6) follows from the continuous-depth limit of the sequential Transformer architecture. This formulation is consistent with a Lie-Trotter splitting scheme, a perspective now well-established in past literature, see [3, 4, 5].
>
> [3] Lu, Y., Li, Z., He, D., Sun, Z., Dong, B., Qin, T., Wang, L., and Liu, T.-Y. Understanding and improving Transformer from a multi-particle dynamic system point of view. In ICLR Workshop: ODE/PDE + DL, 2020.
>
> [4] Geshkovski, B., Rigollet, P., and Ruiz-Balet, D. Measure-to-measure interpolation using transformers, 2024.
>
> [5] Geshkovski, B., Letrouit, C., Polyanskiy, Y., and Rigollet, P. A mathematical perspective on Transformers. Bulletin of the American Mathematical Society, 62(3):427–479, 2025.

---

> > ### Author Rebuttal · Reviewer_ydQ2 · 2026-04-01
> >
> > Thank you for the clarification. I would like to maintain my score.

---

### Official Review · Reviewer_mQqg · 2026-03-12

**Soundness:** 4
**Presentation:** 4
**Significance:** 4
**Originality:** 4
**Overall Recommendation:** 6
**Confidence:** 4

**Summary:**

This paper analyzes the mean-field limit of Transformer token dynamics on ${\mathbb{S}}^{d-1}$, modeled as a Wasserstein gradient flow of an energy functional combining a pairwise exponential interaction (self-attention) with an external potential (feed-forward perceptron). The main finding is that the perceptron qualitatively changes the landscape: while pure repulsive attention admits the uniform measure as a minimizer, adding even a simple ReLU or GeLU potential forces stationary measures to be singular, and generically atomic with finite support. The strongest results hold in $d=2$, where atomicity follows from real-analytic continuation arguments; in general dimensions the paper establishes atomicity via parametric transversality, along with an anti-concentration bound showing no cluster can accumulate too much mass at large inverse temperature.

**Compliance With Llm Reviewing Policy:**

Affirmed.

**Final Justification:**

I maintain my score of 6. The rebuttal answered my questions and I believe this would be a great contribution to mean-field limits of Transformer architectures.

**Key Questions For Authors:**

1. **Convergence to stationary states.** The main theorems characterize stationary measures of the PDE (2.6), but the paper does not establish that solutions starting from smooth initial data (e.g., the uniform distribution) actually converge to these stationary states as $t \to \infty$. The "Takeaway" box on page 2 and the overall framing suggest that the dynamics *produce* atomic configurations, but what is proven is only that *if* the dynamics converge, the limit must be atomic. Can you comment on what conditions are necessary for convergence to hold?

2. **Extension of the anti-concentration bound beyond $d = 2$.** Theorem 3.5 and Corollary 3.6 are stated only for $d = 2$, where the kernel $\theta \mapsto e^{\beta\cos\theta}$ has a clean one-dimensional concavity structure (Lemma A.3). Is there a fundamental obstruction to extending this bound to $d \geq 3$, or is this a technical limitation that you expect to overcome?

3. **What survives when the perceptron drift is non-conservative?** The energy $E_{\beta,\vartheta}$ and its Wasserstein gradient flow structure exist only because the perceptron weight vectors are constrained as $\omega_j = \omega_j a_j$, making the drift a gradient of a scalar potential (eq. (2.7)). In a real Transformer's FFN, the input and output weight matrices are independent, so this symmetry does not hold and no energy functional exists. The notion of "critical points of $E_{\beta,\vartheta}$" then becomes undefined. Do you expect atomicity to persist for general (non-conservative) drifts? Even heuristic arguments would broaden the paper's relevance considerably.

**Limitations:**

Yes

**Strengths And Weaknesses:**

## Strengths

1. **Originality.**
Prior work on the interacting particle / Wasserstein gradient flow formulation of Transformers (Geshkovski et al., 2023; 2025; Chen et al., 2025b; Criscitiello et al., 2024, among others) studied self-attention in isolation. This paper is, to my knowledge, the first to rigorously incorporate the feed-forward (perceptron) block into this framework and characterize its effect on the energy landscape. Since the FFN is a core architectural component present in every Transformer layer, this is a natural and important extension. The key insight is that the perceptron generically forces all stationary measures to be atomic is a strong theoretical contribution that meaningfully advances our understanding of Transformers.

2. **Soundness.**
I checked the proofs and they are correct. The $d=2$ results (Theorems 3.1 and 3.2) make elegant use of the identity theorem for real-analytic functions: the non-analyticity of the ReLU potential forces finitely many zeros of the stationarity condition on each arc, while for analytic activations the argument proceeds through the Wasserstein Hessian to show that infinite support leads to degeneracy, contradicting the strict second-order positive-definiteness assumption. The extension to general $d$ (Theorem 3.3) applies the parametric transversality theorem to establish generic isolation of zeros. The anti-concentration bound (Theorem 3.5) is obtained by a clean computation exploiting the strict concavity of $\theta \mapsto e^{\beta\cos\theta}$ at scale $\beta^{-1/2}$, combined with the second-order condition. I checked the key steps in the appendix proofs (especially A.2 - A.5) and found no errors.

3. **Significance.**
The paper contributes to the general theory of Wasserstein gradient flows with interaction-plus-external energies. The finding that a piecewise-smooth external potential can force singular critical points of a smooth interaction energy is of independent interest beyond the Transformer motivation. The anti-concentration bound and the scaling of heavy atoms with $\sqrt{\beta}$ (Corollary 3.6) add quantitative content beyond a purely qualitative atomicity statement.

4. **Presentation.**
The paper is well-organized and clearly written for a theory-heavy submission. I found the "Model at a glance" box on page 2 to be effective. The distinction between ReLU and analytic activations is maintained consistently throughout, and the hierarchy of results (strongest in $d=2$, weaker but still informative in general $d$) is presented clearly. The simulations, while limited in scope, are well-chosen to illustrate the theoretical predictions: the progression from uniform initialization to atomic clusters is clearly visible in Figures 3, 4, 7.

## Weaknesses

1. **Significance/Soundness.** The gap between the model and real Transformers is substantial and inadequately addressed.
The model assumes (i) $Q^\top K = \beta I_d$ and $V = \pm I_d$ (isotropic attention with identity value matrix), (ii) weights that are constant across layers, (iii) a perceptron drift that is a gradient field, and (iv) unnormalized attention as the primary object. While each simplification is acknowledged, I think their cumulative effect cannot be realized in practice. Real Transformers have layer-dependent, asymmetric key-query matrices and non-conservative feed-forward blocks; I think there is no fixed energy functional whose critical points one can characterize.

2. **Significance.** The strongest results are confined to $d=2$, limiting their relevance. Theorems 3.1, 3.2, and 3.5, and Corollary 3.6 all require $d=2$. In general dimensions, Theorem 3.3 establishes singularity (part (i)) and generic atomicity for analytic activations (part (ii)), but for ReLU (part (iii)) it only guarantees countable atomicity in active regions and says nothing about the "dead zones" where the ReLU potential vanishes. Since practical embedding dimensions are much larger than $d = 2$, the $d=2$ theory, while mathematically clean, has limited direct applicability. Extending the anti-concentration bound beyond $d=2$ would be a substantial improvement; the paper does not discuss whether the concavity-based argument in Theorem 3.5 has a higher-dimensional analogue or faces a fundamental obstruction.

---

> ### Author Rebuttal · Authors · 2026-03-30
>
> **First Question.** Our results characterize stationary measures rather than proving global convergence of the PDE from arbitrary smooth initial data. Exhibiting precise necessary conditions for convergence in this infinite-dimensional setting is generally difficult.
>
> A well-known sufficient condition---which is, in some sense, almost necessary---is to verify a Polyak-Łojasiewicz inequality close to the relevant extrema. In the pure-attention setting, recent work has already established quantitative clustering results from sufficiently regular initializations under suitable assumptions on the parameters; see [1]. This suggests that, at least for sufficiently small MLP parameters, a perturbative local analysis around the attention-only case may be feasible. Establishing such a result rigorously for the coupled attention-MLP model, however, would require substantial additional work and lies beyond the scope of the present paper.
>
> [1] Chen, S., Lin, Z., Polyanskiy, Y., and Rigollet, P. Quantitative clustering in mean-field transformer models, 2025.
>
> **Second Question.** The restriction to $d=2$ is a technical limitation of our current approach rather than a fundamental obstruction. Our proof uses an explicit one-dimensional parametrization of atomic configurations on $\\mathbb S^1$ together with the scalar concavity estimate of Lemma A.3, which yields a transparent finite-dimensional Hessian bound. Extending this argument to $d\\ge 3$ would require controlling the corresponding tangential second-order geometry on small geodesic caps. While similar ideas for handling this higher-dimensional geometry have been successfully leveraged in recent literature (e.g., [1]), adapting these techniques to our specific setting requires significant technical work that falls outside the scope of the present paper. Therefore, we expect the same anti-concentration mechanism to hold in higher dimensions, but leave the formal proof for future work.
>
> **Third Question.** We could possibly expect qualitatively different behavior for general non-conservative drifts. As this closely relates to a point raised by Reviewer 5JoT, we respectfully refer to our detailed response there for a full discussion on this mechanism.

---

> > ### Author Rebuttal · Reviewer_mQqg · 2026-04-04
> >
> > The rebuttal answered my questions. There is no change in my score.

---

### Official Review · Reviewer_psVR · 2026-03-12

**Soundness:** 3
**Presentation:** 3
**Significance:** 2
**Originality:** 2
**Overall Recommendation:** 4
**Confidence:** 3

**Summary:**

The paper studies an existing mean-field spherical model of self-attention and analyzes the effect of adding a perceptron block. Pure self-attention models can lead to equilibrium distributions that are uniformly distributed on the unit sphere $\mathbb{S}^{d-1}$. The main points are:

1. The authors study the Wasserstein gradient flow on a perceptron-modulated energy functional $\mathsf{E}_{\beta,\vartheta}[\mu]$ and argue that the stationary measures are singular and often atomic.
2. In some regimes, atomic stationary measures do not collapse to a single atom.

**Compliance With Llm Reviewing Policy:**

Affirmed.

**Key Questions For Authors:**

Questions and concerns are presented in Strengths and Weakness

**Limitations:**

Yes

**Strengths And Weaknesses:**

**Soundness.**
The paper's main claims seem well-founded and mathematically meaningful. The claims are plausible and reasonably argued, but I do not think all of the results are fully rigorous as written. However, I was convinced of the claims. The mistakes or errors I believe I found do not break the main results.

**Presentation.**
The results are presented coherently and the writing is readable. The overall story is clear and cogent, and I can walk away from the paper with having learnt something. That being said, some important distinctions are not presented in sharp enough resolution. In particular, the paper does not clearly delineate or qualify the main claims with the significant restriction of the perceptron to a limited subclass with special weight symmetries. I found some arguments too compressed.

**Significance.**
The paper presents a genuine non-zero increment of progress in the mean field transformer line of work. The consequence of adding the perceptron layer is convincingly non-trivial and leads to new phenomena. I appreciate the authors for exposing this. However, I'm not overwhelmingly convinced that the model is sufficiently rich or relevant. For example, the weight-symmetry assumption in the perceptron is very restrictive and limits the impact of the work. I understand the assumption is required to ensure appropriate gradient structure, but severely dampens the relevance of the results. I view this paper as an interesting result about a toy transformer system, with weak evidence about practical relevance.

**Originality.**
The paper utilizes a previously-formulated perceptron-modulated energy functional. To my knowledge, the singularity/atomicity of the equilibrium measures are new. The result is incremental, not foundational. The work is justifiably a non-trivial extension of an existing model, not a fundamentally new model.

## Strengths

The paper finds an interesting effect of a perceptron layer in a mean field model of self-attention. The paper makes a generally sound argument that the perceptron layer leads to singular distributions on the sphere. They explore various natural cases to consider.

## Weaknesses

### Theorem clarifications, issues, bugs

**Theorem 3.2**
The proof of Theorem 3.2 uses a standard formula for the Wasserstein Hessian at $\mu$, denoted $\text{Hess}_{\mu} \mathsf{E}_{\beta,\vartheta}$ for $\mu$ in the regular part of $(\mathcal{P}_{2}(\mathbb{S}^{d-1}), W_{2})$ and $\mathsf{E}$ is twice differentiable in the $W_{2}$-sense at such $\mu$. The use of this formula is justified for such $\mu$.

The contradiction hypothesis assumes only that $\text{supp}(\mu)$ is infinite. Does this imply membership in the regular part of $(\mathcal{P}_{2}(\mathbb{S}^{d-1}), W_{2})$? Measures on $\mathbb{S}^{1}$ with infinite support may still be singular. It appears that the proof mixes the validity of the Hessian formula for regular measures with a broader, unspecified notion of regularity for possibly singular measures.

As it stands, the result feels more like a non-existence claim for regular, strictly SOPD critical points than a general classification of stable stationary measures. These issues should be clarified.

Second, there is a confusing statement about test functions (line 880). The authors choose a class of test functions to satisfy the "tangent space constraint" $\int_{}^{} \psi  \, \mathrm{d}\mu=0$. The Wasserstein tangent space is the $L^{2}(\mu)$ closure of gradients. So I don't know what is meant by "tangent space constraint. "

**Theorem 3.3(i)**
The proof says the following:

 > "Exactly as in the last part of the proof of Theorem 3.1, combining these relations for adjacent cells forces $v_{\vartheta}$ to be a quadratic polynomial on all of $\mathbb{S}^{d-1}$, hence real-analytic."

I fear this compresses some non-trivial details. The present higher-dimensional case involves cells separated by codimension-1 faces of hyperplanes. I'd like to see this higher-dimensional "gluing" argument fleshed out. This step is asserted, but not proven.


**Theorem 3.3(ii)**
The *general* perceptron drift was defined (Lines 173-175)
$$
\mathsf{u}_{\vartheta}(x) = P_{x}^{\perp} \sum_{j=1}^{K} \boldsymbol{\omega}_{j} \sigma(a_{j} \cdot x)
$$
where $a_{j} \in \mathbb{R}^{d}, \boldsymbol{\omega}_{j} \in \mathbb{R}^{d}$. Each unit has $2d$ real parameters and with $K=d$ units the parameters space is $(\mathbb{R}^{2d})^{d}$. Theorem 3.3(ii) is stated over a parameter set $U_{\mu} \subset \mathbb{R}_{>0} \times (\mathbb{R}^{2d})^{d}$ which appears to correspond to the *general drift* above. The proof in Appendix A.3 Part II is valid only for the restricted case where parameter symmetries $\boldsymbol{\omega}_{j} = \omega_{j} a_{j}$ hold. Such a perceptron has only
$$
d(d+1) = d^{2} + d < 2d^{2}
$$
parameters. The theorem statement and its proof appear to live in different parameter spaces — unclear to me if this is a typo in the statement or something more problematic.

**Theorem 3.3(iii)**
The statement of the result also shares the same problem as the previous.

### Errors

 **Lemma A.3**
 concavity threshold angle is given as
$$
\theta_{c}(\beta) = \arccos \left( \frac{\sqrt{ 1+4 \beta^{2} }-1}{2\beta} \right).
$$
They claim that for small $\beta$, $\theta_{c}(\beta) \sim \pi/4$. But as $\beta \to 0$,
$$
\frac{\sqrt{ 1+4\beta^{2} }-1}{2\beta} \approx \beta.
$$
This implies
$$
\theta_{c}(\beta) \approx \arccos(\beta) \to \frac{\pi}{2}
$$
and not $\pi / 4$.

**Remark A.2**
The injectivity argument is invalid. Assume $f_{B,\nu}(x) =0$ for all $x \in \mathbb{S}^{d-1}$. Then the argument claims the following: because $B$ is non-singular, for every $z \in \mathbb{R}^{d}$there is a $t > 0$ and $x \in \mathbb{S}^{d-1}$ such that $z = t B^\top x$, therefore
$$
F(z):=\int_{}^{} \mathrm{e}^{ z \cdot y } \, \mathrm{d}\nu(y) = 0.
$$
This does not follow. The hypothesis $f_{B,\nu}(x)=0$ is only the statement that
$$
F(B^\top  x) = 0 \qquad \forall \, x \in \mathbb{S}^{d-1}
$$
which is only the special case where $t=1$ in $z = t B^\top x$. This gives no information about $F(t B^\top x)$ for $t \neq 1$.

I think the claim is true as written, but I don't think the proof is correct.

---

> ### Author Rebuttal · Authors · 2026-03-30
>
> ### Theorem clarifications, issues, and bugs
>
> **Theorem 3.2.** We agree that the previous phrasing was too broad. We have clarified that this result is not a complete classification of all stable measures, but rather rules out infinite-support strictly SOPD critical points among measures with a well-defined second variation.
>
> Regarding the tangent space, the condition $\\int \\xi\\,\\mathrm{d}\\mu = 0$ is not intended to characterize the Otto tangent space. Rather, $\\xi$ is a tangent perturbation, and in the proof the zero-mean condition is used only to subtract the constant $\\mathsf{K}_\\beta''(0)$ in the bilinear form. We now state this explicitly.
>
> **Theorem 3.3(i).** In the revised argument, this step is avoided altogether. Once one cell $I$ with $\\sigma_d(\\operatorname{supp}(\\mu) \\cap I) > 0$ is found, analyticity implies
>
> $$
> \\frac{\\delta \\mathsf{E}\_\\beta}{\\delta \\mu}[\\mu]=C_I - \\frac{1}{2}\\sum_{j\\in J_I}\\omega_j(a_j\\cdot x)^2\\qquad \\text{on } \\mathbb{S}^{d-1}.
> $$
>
> By Lemma A.1(i), $\\mu$ then has a polynomial density, hence full support. The stationarity condition therefore holds on all of $\\mathbb{S}^{d-1}$, so
> $$
> \\frac{\\delta \\mathsf{E}\_\\beta}{\\delta \\mu}[\\mu] + \\frac{1}{2}\\mathsf{v}\_\\vartheta\\qquad \\text{is constant on }\\mathbb{S}^{d-1}.
> $$
> Since $\\frac{\\delta \\mathsf{E}\_\\beta}{\\delta \\mu}[\\mu]$ is already globally quadratic, it follows directly that $\\mathsf{v}_\\vartheta$ is globally quadratic, hence real-analytic, contradicting the assumption.
>
> **Theorems 3.3(ii) and 3.3(iii).** The reviewer is correct: there was a mismatch in parameter space. The theorem was inadvertently stated for the general perceptron, but the proof requires $\\boldsymbol{\\omega}\_j = \\omega\_j a\_j$ in order to define $\\mathsf{E}\_{\\beta,\\vartheta}$. We have corrected this inconsistency in the statement, and both theorems now use parameter sets
> $$
> U(\\mu) \\subset (0,\\infty) \\times (\\mathbb{R}^{1+d})^d.
> $$
>
> ### Errors
>
> **Lemma A.3.** The reviewer is right. For small $\\beta$, the expansion is
> $$
> \\frac{\\sqrt{1+4\\beta^2}-1}{2\\beta} = \\beta + O(\\beta^3).
> $$
> Consequently, $\\theta\_c(\\beta) \\to \\pi/2$ as $\\beta \\to 0^+$, rather than $\\pi/4$. We have corrected this in the revised manuscript. Importantly, this does not affect any subsequent proof or result.
>
> **Remark A.2.** We thank the reviewer for spotting this flaw. The reviewer is correct that knowing $F(B^\\top x) = 0$ for $x \\in \\mathbb{S}^{d-1}$ only provides information on the surface of an ellipsoid, so the extrapolation to $t \\neq 1$ in the previous proof was invalid.
>
> The claim itself remains true, and we have replaced the proof in the revised manuscript with a new argument. Define $\\Omega$ to be the solid region bounded by
> $$
> \\partial\\Omega = \\{B^\\top x : \\|x\\| = 1\\}.
> $$
> The hypothesis implies that
> $$
> F(z) \\coloneqq \\int e^{z\\cdot y}\\,\\mathrm{d}\\nu(y) = 0
> \\qquad \\text{on } \\partial\\Omega.
> $$
> Moreover, since $\\nu$ is supported on $\\mathbb{S}^{d-1}$, differentiation gives
> $$
> \\Delta F(z) = \\int \\|y\\|^2 e^{z\\cdot y}\\,\\mathrm{d}\\nu(y) = F(z).
> $$
> Hence $(\\Delta - 1)F = 0$ on $\\mathbb{R}^d$, and uniqueness for the Dirichlet problem on $\\Omega$ implies that $F \\equiv 0$ in $\\Omega$. Real-analyticity then yields $F \\equiv 0$ on all of $\\mathbb{R}^d$, from which injectivity follows.

---

> > ### Author Rebuttal · Reviewer_psVR · 2026-04-04
> >
> > My concerns have been addressed.

---

### Official Review · Reviewer_5JoT · 2026-03-15

**Soundness:** 4
**Presentation:** 4
**Significance:** 4
**Originality:** 4
**Overall Recommendation:** 6
**Confidence:** 5

**Summary:**

This paper studies the representations dynamics of deep Transformer architectures by modeling token embeddings as a mean-field interacting particle system on the sphere and analyzing the associated Wasserstein gradient flow of an interaction energy. Building on recent work that characterizes the clustering behavior of pure self-attention dynamics, the paper provides an essential contribution to the analysis of the models, by integrating the key feed-forward perceptron block into the mean-field formulation, formalized as an additional drift term derived from a neural network potential. Furthermore, the paper provides a detailed characterization of stationary points of the resulting energy landscape. In particular, the authors show that, generically, stationary measures are singular and typically purely atomic, implying that the presence of MLP layers tends to produce clustered steady states even in regimes where attention-only models admit diffuse equilibria. The paper further quantifies the structure of these clusters, including quantitative bounds on cluster masses and on the number of atoms as a function of the interaction scale. Well-designed numerical simulations illustrate the emergence of clustering in practice, confirming the theoretical predictions.

**Compliance With Llm Reviewing Policy:**

Affirmed.

**Key Questions For Authors:**

- To what extent do the atomicity and clustering results persist under more general attention kernels or anisotropic weight matrices?
- Out of curiosity, do the authors expect that some "problematic" dynamical behavior could happen when their assumptions on the MLP structure fail to hold?
- do the authors expect the occurrence of phase transitions for the maximizers of the energy towards the anti-concentrated regime as $\beta$ increases?

**Limitations:**

yes, the limitations of the work are addressed clearly by the authors

**Strengths And Weaknesses:**

**Significance and originality**: The paper addresses one of the central unresolved problems in the mathematical theory of Transformers, namely how to incorporate the MLP/perceptron block into the theoretical investigation of the model's representation dynamics. In that sense, the work fills what has arguably been the main conceptual gap in this rapidly developing line of research: it finally integrates the MLP layer into the theory, thereby bringing the mathematical analysis substantially closer to the actual architecture of these models. Moreover, the paper extracts a groundbreaking conceptual insight about the residual-stream dynamics of Transformers, showing that the clustering phenomenon identified in previous literature is in fact *universal*, in the sense that it arises generically also in the full coupled architecture, with the perceptron actively selecting and enforcing such localization.  In this sense, this paper provides a major advance in our understanding of these models and constitutes a milestone contribution to the field.

**Soundness and presentation**: The results are mathematically precise and technically strong. Theorems such as the finite-support characterization of stationary measures in low dimensions and the generic atomicity results in higher dimensions provide a clear structural description of the landscape. The anti-concentration bounds further quantify the mass distribution across clusters. The work substantially advances the theoretical understanding of Transformers by introducing what appears to be the first framework that incorporates intrinsic neural network components into the mean-field landscape analysis of these models. Numerical experiments are carefully designed and illustrative. They convincingly demonstrate the emergence of clusters and the predicted scaling behavior with respect to the interaction parameter, and they help bridge the gap between the abstract analysis and concrete dynamics. The paper is clearly written and well organized. The results are precisely stated, and the work is well contextualized within the recent literature on mean-field analyses of attention dynamics.

In summary, this is a groundbreaking paper that addresses an essential and presently unsolved problem in the theory of large language models, it is very well written, it provides strong mathematical results supported by convincing experiments. In my view, this paper constitutes one of the main contributions on the theoretical study of transformers in the last few years.

---

> ### Author Rebuttal · Authors · 2026-03-30
>
> **First Question.** Our results are robust under anisotropic bilinear attention, provided the interaction is of the form $e^{x^\top B y}$ with $B$ symmetric and invertible. This extension is already discussed in Remark 3.4 (see also Remarks A.2 and B.2). The role of symmetry is important here, since it is what allows one to keep the interaction in the Wasserstein gradient-flow framework. For a merely invertible but non-symmetric $B$, this variational structure is lost, so our stationary analysis does not directly apply in that setting.
>
> For more general attention kernels $K$, the picture depends on which part of our analysis one has in mind. For the atomicity/singularity results, what really matters is that the interaction transform
> $$
> \mu \mapsto \left(x \mapsto \int K(x,y)\\,\mathrm{d} \mu(y)\right)
> $$
> remain real-analytic in $x$ and injective in $\mu$, or with a comparable substitute for this injectivity mechanism. Under such assumptions, we expect the analyticity-based part of the argument to extend beyond the bilinear kernel.
>
> By contrast, the clustering/anti-concentration bounds use a local concavity estimate near the diagonal (Lemma A.3) and a corresponding Hessian control. So for completely general kernels, extending those bounds would require a quantitative replacement for this concavity mechanism. We therefore expect some robustness beyond $e^{\beta x\cdot y}$, but only for kernels with analogous analytic and local-curvature properties.
>
> **Second Question.** We could expect qualitatively different behavior once the structural assumptions on the MLP fail. When the structural assumptions on the MLP fail, the velocity field $v$ is no longer a gradient. Long-time attractors satisfy the general stationary condition $\operatorname{div}(\mu v) = 0$. In our symmetric model, $v$ is a gradient, which forces this condition to simplify to the pointwise equality $v = 0$ on $\operatorname{supp}(\mu)$, as stated in (2.8).
>
> We do not claim that the clustering mechanism fails for general, non-conservative drifts. However, our proofs rely strictly on the pointwise equation $v = 0$ on $\operatorname{supp}(\mu)$ as their starting point. Handling the general condition $\operatorname{div}(\mu v) = 0$ falls outside the scope of our current techniques and remains an open problem.
>
> **Third Question.** Interpreted literally in terms of global maximizers, our model does not exhibit a $\beta$-driven transition toward an anti-concentrated regime: by Proposition 3.7(i), the global maximizers of $\mathsf{E}\_{\beta,\theta}$ are always Dirac masses, so they remain fully concentrated for every $\beta\in(0,\infty)$. The reason the anti-concentration bound does not apply to these points is that they are not SOPD.
>
> That said, we believe your question is very natural when interpreted in terms of global minimizers in the repulsive (descent) regime. There, for large $\beta$, our anti-concentration bound shows that SOPD critical points cannot place too much mass inside a ball of radius $O(\beta^{-1/2})$. We do not analyze this possible transition in the present paper, but we agree that understanding the $\beta$-dependence of minimizers and stable states is a very interesting direction for future work.

---

> > ### Author Rebuttal · Reviewer_5JoT · 2026-04-03
> >
> > The authors have satisfactorily addressed my questions. I will keep my positive score unchanged.

---

### Decision · Program_Chairs · 2026-04-30

**Decision:**

Accept (spotlight)

**Comment:**

The authors study a toy model of an infinite depth transformer, which for the first time also contains an FFN block into the mean field model. The reviewers found the main result interesting: the FFN block leads to singular equilibria. In my opinion, it's not particularly surprising, but it's nice to have it well formulated and handled rigorously.

The review process discussion two types of main criticisms:
1. The model remains too idealized, and not that close to a realistic transformer.

I tend to agree with this sentiment, albeit the reviewers and I don't find this a critical issue. It does not help me raise this paper towards a higher recommendation, but also I don't see this as a reason to reject either.

2. Several technical issues were raised regarding precision and gaps in proofs.

These were very well addressed by the authors, and the reviewers tend to agree that they are considered resolved.

Overall, I believe this paper contains non-trivial theoretical contributions, which are now precisely handled, despite still remaining in a somewhat idealized setting. I would strongly suggest accept.